EMBO
Molecular Medicine

# Replicated blood-based biomarkers for myalgic encephalomyelitis not explicable by inactivity

Sjoerd Viktor Beentjes [ID][1,2,3,6 ✉], Artur Miralles Méharon [ID][4], Julia Kaczmarczyk[1], Amanda Cassar[4], Gemma Louise Samms[2], Nima S Hejazi[5], Ava Khamseh [ID][2,3,4,6 ✉] & Chris P Ponting [ID][2,6 ✉]

## Abstract

Myalgic encephalomyelitis/chronic fatigue syndrome (ME/CFS) is a common female-biased disease. ME/CFS diagnosis is hindered by the absence of biomarkers that are unaffected by patients' low physical activity level. Our analysis used semi-parametric efficient estimators, an initial Super Learner fit followed by a one-step correction, three mediators, and natural direct and indirect estimands, to decompose the average effect of ME/CFS status on molecular and cellular traits. For this, we used UK Biobank data for up to 1455 ME/CFS cases and 131,303 controls. Hundreds of traits differed significantly between cases and controls, including 116 significant for both female and male cohorts. These were indicative of chronic inflammation, insulin resistance and liver disease. Nine of 14 traits were replicated in the smaller All-of-Us cohort. Results cannot be explained by restricted activity: via an activity mediator, ME/CFS status significantly affected only 1 of 3237 traits. Individuals with post-exertional malaise show stronger biomarker differences. Single traits could not cleanly distinguish cases from controls. Nevertheless, these results keep alive the future ambition of a blood-based biomarker panel for accurate ME/CFS diagnosis.

**Keywords** Myalgic Encephalomyelitis; Semi-parametric Mediation Analysis; Post-exertional Malaise; UK Biobank; All of Us Biobank
**Subject Categories** Biomarkers; Neuroscience

## Introduction

Myalgic encephalomyelitis (ME; also known as chronic fatigue syndrome, CFS) is an often debilitating disease of unknown pathogenesis defined by post-exertional malaise (PEM), the dramatic worsening of symptoms after even minor mental or physical exertion (Committee on the Diagnostic Criteria for Myalgic Encephalomyelitis/Chronic Fatigue Syndrome, 2015), which usually persists at least 24 h, in contrast to other fatiguing illnesses (Cotler et al, 2018). ME/CFS has no cure and no widely effective therapy (National Institute for Health and Care Excellence (NICE), 2021). It is a female-dominant disease (Bretherick et al, 2023; Jason et al, 2022). There are no clinical biomarkers for ME/CFS. A high priority for people with ME/CFS is an accurate and reliable diagnostic test (Tyson et al, 2022). Findings from dozens of biomarker studies have shown limited reproducibility, perhaps due to their typically low sample sizes, their frequent use of inappropriate statistical tests (Maksoud et al, 2023) and the known heterogeneity of ME's symptoms and potential aetiology (Huber et al, 2018).

Any clinical biomarker would need to account for individuals' inactivity relative to the general population. This is because many people with ME/CFS do not exercise and have to restrict their activity (Silver et al, 2002) to reduce the risk of subsequent PEM. Some have proposed that it is this avoidance of activity that inhibits recovery by perpetuating ME/CFS symptoms following an acute illness (Wessely et al, 1989; Moss-Morris et al, 2013; Sharpe, 1995). According to this theory, a gradual return to activity reduces fatigue and disability by reversing deconditioning (White et al, 2011) and would reverse any physiological changes, for example in blood traits, caused by inactivity (White, 2000). Counter to this theory, however, is that therapies based on physical activity or exercise are ineffective as a cure (National Institute for Health and Care Excellence (NICE), 2021), implying that ME/CFS is instead an ongoing organic illness (Geraghty et al, 2019; Committee on the Diagnostic Criteria for Myalgic Encephalomyelitis/Chronic Fatigue Syndrome; Board on the Health of Select Populations; Institute of Medicine, 2015).

In this study, we analysed data from the UK Biobank (UKB), a population cohort of 500,000 individuals aged between 40 and 69 at recruitment linked to diverse phenotypic data (Bycroft et al, 2018). We analysed molecular and cellular data from up to 1455 UKB participants, all of whom self-reported a clinical diagnosis of CFS and/or ME/CFS, and from up to 131,303 controls. From other studies, we expect approximately half of these cases to conform to the Canadian consensus criteria (Newton et al, 2010; Nacul et al, 2011; Devasahayam et al, 2012). These criteria require PEM, and indeed, among cases completing the Pain Questionnaire, 55% report PEM-like symptoms. This allowed our final analysis to

---

[1]School of Mathematics and Maxwell Institute for Mathematical Sciences, University of Edinburgh, Edinburgh EH9 3FD, UK. [2]MRC Human Genetics Unit, Institute of Genetics & Cancer, University of Edinburgh, Edinburgh EH4 2XU, UK. [3]Division of Biostatistics, University of California, Berkeley, CA 94720-7360, USA. [4]School of Informatics, University of Edinburgh, Edinburgh EH8 9AB, UK. [5]Harvard T.H. Chan School of Public Health, Department of Biostatistics, Boston, MA 02115, USA. [6]These authors contributed equally: Sjoerd Viktor Beentjes, Ava Khamseh, Chris P Ponting. ✉E-mail: Sjoerd.Beentjes@ed.ac.uk; ava.khamseh@ed.ac.uk; Chris.Ponting@ed.ac.uk

**Table 1. Numbers of UKB ME/CFS cases or non-ME/CFS controls per category.**

|          | Blood traits | | NMR metabolites | | Proteomics | |
|----------|-------|----------|-------|----------|-------|----------|
|          | Cases | Controls | Cases | Controls | Cases | Controls |
| Female   | 1069  | 75,731   | 615   | 41,436   | 126   | 7963     |
| Male     | 386   | 55,572   | 213   | 30,921   | 45    | 5920     |
| Combined | 1455  | 131,303  | 828   | 72,357   | 171   | 13,883   |

compare the effect of ME/CFS status on blood traits for those with and without PEM.

Our three groups of analyses were on (i) 31 blood cell and 30 blood biochemistry phenotypes; (ii) 251 NMR-measured metabolites; and (iii) 2923 proteins. Specifically, we quantify which blood traits, Nuclear Magnetic Resonance (NMR) metabolomics, and proteomics features are significantly different between ME/CFS cases versus controls, controlling for age and sex. The large UK Biobank datasets for ME/CFS cases and controls provided substantial statistical power to evaluate hypotheses, also allowing comparison between male-only and female-only analyses, something that had not been previously achievable. We take advantage of three mediators of sedentary lifestyle to determine whether any molecular or cellular trait associated with ME/CFS cases is explicable by physical inactivity.

# Results

## Study population: cases and controls

We first defined 1455 ME/CFS cases and 131,303 non-ME/CFS population control individuals from the UKB (Bycroft et al, 2018). No single item of the UKB questionnaire or electronic health record data provides incontrovertible evidence of ME/CFS status. Consequently, our choice of case definition was guided by a large fraction (64%) of those self-reporting a CFS diagnosis at baseline having one or more further pieces of evidence (Pain Questionnaire ME/CFS response, or ME/CFS-related codes (Dataset EV1) in General Practice or Hospital Episode Statistics data) in support of their ME/CFS status, despite data missingness (Samms and Ponting, 2025b). Cases reported their clinical diagnosis of CFS or ME/CFS at least once and poor or fair overall health; controls had no evidence for a ME/CFS diagnosis and were of good or excellent health (see "Methods"). For each group of analyses, cases and controls were restricted to those with measurements of 31 blood count and 30 blood biochemistry markers, or 251 NMR metabolites, or 2923 protein levels, respectively. Collection of these biological samples was contemporaneous with self-reporting of CFS at the first visit to a UKB Assessment Centre (2006–2010). Numbers of samples in each category are shown in Table 1. ME sample sizes for measured outcome in blood traits, NMR metabolites and proteins are shown in Fig. EV1.

## Molecular and cellular traits significantly associated with ME/CFS

We simultaneously quantified two effects of ME/CFS case status on molecular and cellular traits, the natural direct effect (NDE) and

natural indirect effect (NIE) (Fig. 1A). Each measures the difference between averages for ME/CFS cases or controls, weighted by activity level and also correcting for age and sex because levels of some molecules are age- (e.g., HRG protein) and/or sex-dependent (e.g., ALT). NDE and NIE account for the activity mediator by decomposing the average effect of ME/CFS case status on molecular or cellular trait into (a) direct paths—those not involving the mediator (Fig. 1A, green)—and (b) indirect paths—those acting through the mediator (blue)—with level of activity as the mediator variable (Robins and Greenland, 1992; Pearl, 2001) (see "Methods").

As a mediator variable, we first used "Duration of walk" (UKB field 874). As expected, ME/CFS cases reported a lower duration of walk (mean: 44.0 min/day) than controls (55.3 min/day). At a false discovery rate (Benjamini and Hochberg, 1995) (FDR) <0.05, significant direct effects were found for 36 (of 61 + 2 composite) blood traits, 189 (of 251) NMR metabolites and 65 (of 2923) proteins (Fig. 1A). All estimates restrict to complete cases, removing individuals with missing trait data in that estimate. For all three analyses, the number of significant NDE results and their intersection in each of the male, female and combined categories are presented in Fig. 1B.

Significant NDEs on molecular and cellular blood traits for females or males or combined are shown in Fig. 2A. NDEs are strongly correlated between females and males Fig. 2B. Twenty traits are separately significant in the two sexes (Fig. 2A,B) and thus their associations to ME/CFS status are independently replicated. In addition, a single trait (erythrocyte_distribution_width, sometimes a sign of anaemia) was also significant with positive NDE for males and negative NDE for females (Fig. 2A).

Among the 20 significantly associated traits for females and for males were traits indicative of chronic inflammation (elevated C-reactive protein [CRP] and cystatin C levels, and leucocyte and neutrophil counts), insulin resistance (elevated triglycerides-to-HDL cholesterol [TG-to-HDL-C] ratio, alanine aminotransferase [ALT], alkaline phosphatase [ALP] and gamma glutamyltransferase [GGT]), and liver disease (elevated ALT, ALP and GGT, and low urea levels) (Fig. 2A). The full results can be found in Datasets EV2 and EV3, with the case and control numbers of all analyses in Dataset EV4. Figure 2C illustrates the shifts in two measures of insulin resistance, the TyG index (Fritz et al, 2019; Won et al, 2020) (top) and TG-to-HDL-C ratio (Oliveri et al, 2024) (bottom), between ME/CFS cases and controls. These are the UKB raw data, rather than results from mediation analysis.

## Physical activity does not explain ME/CFS associations to blood traits

Strikingly, for the UKB 874 mediator, significant effects on ME/CFS case status were abundant for direct effects (i.e., NDE; Figs. 2A and EV2), but occurred only once (mean_corpuscular_haemoglobin; adjusted $P = 0.043$) for indirect effects (i.e., NIE; Fig. 3). For every other one of the 61 + 2 composite blood traits, for females or males or both sexes combined, none was significant when controlling the FDR at ≤0.05 (Fig. 3). Results from applying two other mediators (Figs. 2A and 3) are presented later.

## Metabolite traits significantly associated with ME/CFS

Of 251 NMR metabolite traits, 189 (75%) were significantly associated with ME/CFS status in an NDE analysis with females

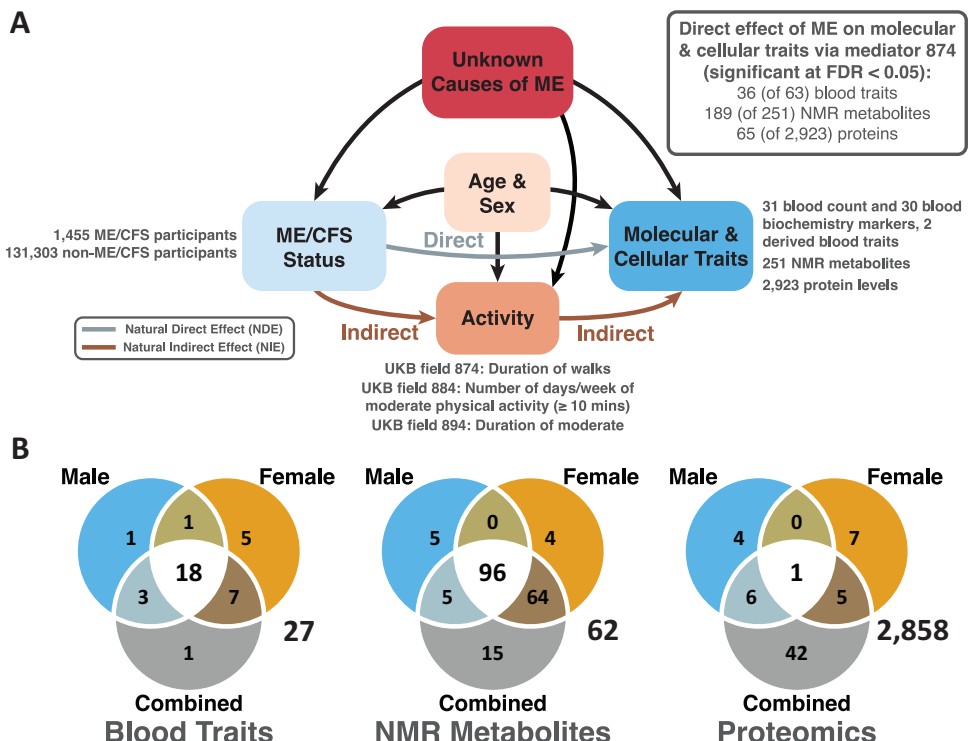

**Figure 1. Study design and overview of results.**

(A) Directed Acyclic Graph for ME/CFS, taking age and sex as confounders and sedentary lifestyle (physical activity) as a mediator for ME/CFS's effect on molecular and cellular traits. The causes of ME/CFS are an unknown variable (red). Therefore, all effect estimators are quantifying an association between ME/CFS and molecular or cellular traits and no causal statements are made. The "Age" variable (UKB field 21022) represents age at recruitment to UKB, rather than age of onset or diagnosis of ME/CFS. This variable affects the probability of having a ME/CFS diagnosis: recovery is minimal (≈5%, (Cairns and Hotopf, 2005)), and as they age people are increasingly likely to be diagnosed with ME/CFS. As it also affects the molecular and cellular traits, age is treated as a confounder. (B) Venn diagrams displaying the number of significant findings in the male, female, and combined cohorts, and their intersection for Natural Direct Effect (NDE, grey), mediator 874. Proteomics data have the smallest sample size (see Table 1) and least power, implying fewer significant results in males and females separately as compared to the combined analysis.

only (68 traits) or males only (10 traits) or in both the females only and males only analyses (96 traits) (UKB 874 mediator; Fig. 1B; Dataset EV5). Significant traits were mostly lipid levels, involving lipoproteins, cholesterol, and triglycerides. Results were highly concordant between females only and males only analyses (Fig. 4A,B) indicating that ME/CFS-specific blood metabolite differences are, again, generally not sex-biased. Previous ME/CFS metabolomic biomarker studies used one and three orders-of-magnitude fewer cases and controls, respectively (Maksoud et al, 2023). The largest among these identified lowered phosphatidyl-cholines and cholines in blood from ME/CFS cases ((Nagy-Szakal et al, 2018), see also (Naviaux et al, 2016)), results that we replicated here (Fig. 4A). Higher triglycerides and lower HDL cholesterol in ME/CFS cases, observed using UKB enzymatic assays (Fig. 2A), were also observed as significant in the NMR metabolomics assays (Fig. 4A). Of nine amino acids measured, only alanine was significantly elevated, and then only in female ME/CFS cases. Blood pyruvate and lactate, previously predicted to be ME/CFS biomarkers (Yamano et al, 2016; Ghali et al, 2019), were also not significantly different between cases and controls.

None of the 251 metabolite traits was significant when controlling the FDR at ≤0.05 for indirect effects using the "Duration of walk"' (UKB 874) mediator, for females or for males or for both combined (Fig. EV2B).

## Proteomic traits significantly associated with ME/CFS

Repeating this NDE analysis using the UKB 874 mediator on levels of 2923 proteins, measured using antibody-based assays, yielded only a single protein, extracellular superoxide dismutase or SOD3, whose abundance was significantly altered (FDR <0.05) between cases and controls in both females and males. Relative to preceding analyses, this proteomic analysis is under-powered owing to there being fewer cases for whom data were available (Table 1) and its larger multiple testing burden. Implications of this association to SOD3 are unclear, although superoxide, SOD3's substrate, is known to modulate the hyperalgesic response (Wang et al, 2004).

Male- or female-specific effects for the same protein are again correlated (Fig. 5; Dataset EV6). Considering all cases combined, 54 proteins are significant (FDR <0.05; Fig. 1B). Among these are 7 complement proteins (C1RL, C2, CFB, CFH, CFI, CFP and CR2) of the innate immune system, whose levels are all elevated in ME/CFS cases, including CR2 (complement C3d receptor 2), the receptor for Epstein–Barr virus (EBV) binding on B and T lymphocytes. Two of the up-regulated proteins (CDHR2 and CDHR5) together form the extracellular portion of the intermicrovillar adhesion complex, whose disruption leads to intestinal dysfunction and inflammatory bowel disease (Crawley et al, 2014; Mödl et al, 2023). ME/CFS cases also show an increase in levels of leptin (LEP), which has a role in

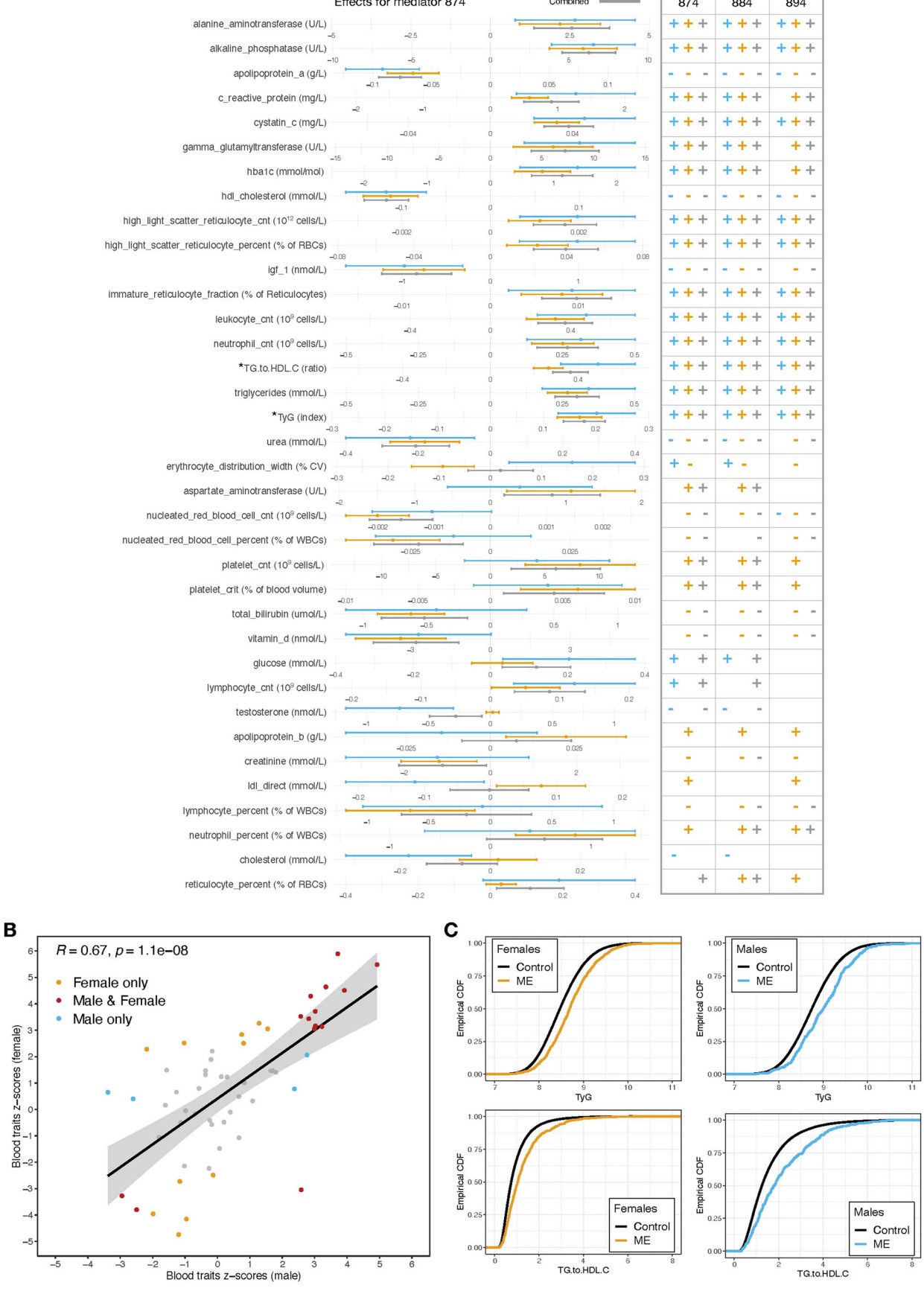

**Figure 2. Associational natural direct effects (NDE) of ME/CFS on molecular and cellular blood traits.**

(A) The sex-stratified analyses are presented in orange (female) and blue (male). For the combined analysis (grey), sex is additionally taken as a confounder. All traits that are significant for the UKB 874 mediator are shown (see Dataset EV3 for the UKB 884 and 894 mediators). Natural direct effect sizes (left) are plotted for the UKB 874 mediator ("Duration of walks"), for significant estimates (FDR <0.05). Error bars indicate 95% confidence intervals, and the central point represents the population average estimate. Note that the scale and unit of measurement for each trait (x axis) are different. For example, the unit of measurement of alanine aminotransferase (Field 30620) is U/L. The analysis was repeated for the UKB 884 mediator ("Number of days/week of moderate physical activity") and for the UKB 894 mediator ("Duration of moderate activity"), with the significant results (FDR <0.05) in each category indicated by "+"' symbols for positive effects and "−" for negative effects (right). Where there is no symbol, the effect was not significant. Notably, there were no discordant results across the three mediators. All blood trait names are as they appear in the UKB showcase, aside from TyG and TG-to-HDL-C ratio (indicated by *), which are composite measures of other blood traits. Full results can be found in Datasets EV2 and EV3. Sample sizes of each analysis are in Dataset EV4. (B) Blood trait NDE z-scores, males (x axis), females (y axis). Z-scores are the NDE divided by its estimation error. The Pearson correlation is 0.67 and significant. The red dots represent 14 blood traits that are significant in both males and females (FDR <0.05). The yellow and blue dots represent blood traits that are significant in females only and males only, respectively (FDR <0.05). The grey dots are significant in neither group while controlling FDR <0.05. (C) Raw data empirical cumulative distribution functions (ECDFs) for TyG (top) and TG-to-HDL-C ratio (bottom), comparing controls (black) and cases (female on the left, male on the right).

energy homeostasis (Triantafyllou et al, 2016). Again, not a single protein among the 2923 yielded a significant NIE estimate for this mediator (Fig. EV2B).

## Total effects

We have shown above that direct effects dominate, so that indirect effects contribute little or nothing to molecular and cellular effects. In real-world settings, the quantity of most interest to clinicians will be the total effect (TE), accounting for age and sex, rather than the direct effect. Estimating the total effect for 63 blood traits finds 39 to be significant (FDR <0.05) predictors of ME/CFS case status for females and males combined (Fig. EV2A). The traits that are robustly predictive of ME/CFS are those shown in Fig. 2A (with four exceptions: *erythrocyte_distribution_width*, *apolipoprotein_b*, *creatinine* and *ldl_direct*). Blood trait results calculated for females or males separately are highly correlated (Fig. 6A). For one or more of female- or male-specific or combined TE analyses, a total of 251 proteins and 216 metabolites were additionally significant (FDR <0.05; Figs. 6B,C and EV2A).

Significantly enriched Gene Ontology (GO) terms for TE-significant proteins highlighted tumour necrosis factor (TNF) and interleukin-4 (IL4) production, and natural killer (NK) cell-mediated cytotoxicity (Fig. EV3). Nevertheless, TNF and IL4 proteins themselves were not significantly altered in abundance. Impaired NK cell cytotoxicity in ME/CFS, however, is one of the few cellular or molecular biomarkers that has often been replicated (Eaton-Fitch et al, 2019).

## Sensitivity analyses for blood traits

Next, we investigated whether blood trait results replicate for two further mediators: "Number of days/week of moderate physical activity 10+ min" (UKB field 884) and "Duration of moderate activity" (UKB field 894) questionnaire responses. As expected, ME/CFS cases reported less activity than controls: mean 2.77 vs 3.51 days/week, and 53.9 vs 60.0 min/day for mediators 884 and 894, respectively. As before, significant effects on ME/CFS status were observed for direct effects, never indirect effects for the "Duration of moderate activity" mediator (UKB field 894) (Figs. 2A and 4). By contrast, for the "Number of days/week of moderate physical activity 10+ min" (UKB field 884) mediator, 22 significant NIEs were identified: 14 (0.4%) and 8 (0.2%) traits at FDR ≤0.05 for combined female and male, and

female-only data, respectively. Importantly, even when significant NIEs are found, they almost always contribute less to the total effect than NDEs (Fig. 6D). To obtain a high confidence set, we have included a restricted set of traits significant for NDE for females and for males (mediators 874 and 884) and for females (mediator 894), resulting in 18 traits listed in Dataset EV7.

We additionally investigated the dependence of results on the choice of fitting algorithm(s) for blood traits. Specifically, the results in Fig. 2A were obtained using a cross-validating library of algorithms (SuperLearner (SL), see "Methods"). Results obtained with no SL—reducing the library to the baseline GLMnet—with mediator 874, for TE, NDE and NIE are provided in Dataset EV8. Although we recommend its use, leaving out the SL has only minor effect: 36 of 39 significant TE blood traits using UKB field 874 as mediator with the SL were also significant without its use, Dataset EV8. Full NDE and NIE values for all mediators with SL are provided in Datasets EV2 and EV3.

For TEs, 41 blood traits (as well as TyG and TG-to-HDL-C ratio) differ significantly between female or male ME/CFS cases and controls (Dataset EV2). To test whether extreme values affect these results, we winsorized the blood trait data at 0.5 and 1%. The results on the combined dataset are presented in Fig. EV4 and Dataset EV9, and remain robust.

Lastly, we found that TEs and NDEs increase as the stringency of case and control definitions increases (Fig. 7; see Dataset EV10 for full results). Specifically, we compared NDEs for molecular and cellular blood traits calculated from cases and controls as defined in "Methods", but with or without overall health rating (UKB field 2178) of 'Poor' or 'Fair' at baseline for cases, and/or 'Good' or 'Excellent' for controls. Removing general health criteria when defining cases and controls did not substantially affect our statistically significant discoveries (Fig. 7).

## More biomarkers are significant for those with PEM than for those without PEM

Among the ME/CFS cases with available Pain Questionnaire data, 55% reported long-lasting PEM-like symptoms ("Methods"). This allowed us to compare blood trait results for those with PEM versus those without PEM symptoms. For these two analyses, we used disjoint sets of controls and equal numbers of cases to match statistical power. Results from the two analyses were highly concordant (Fig. 8A). Nevertheless, whereas 26 biomarkers were

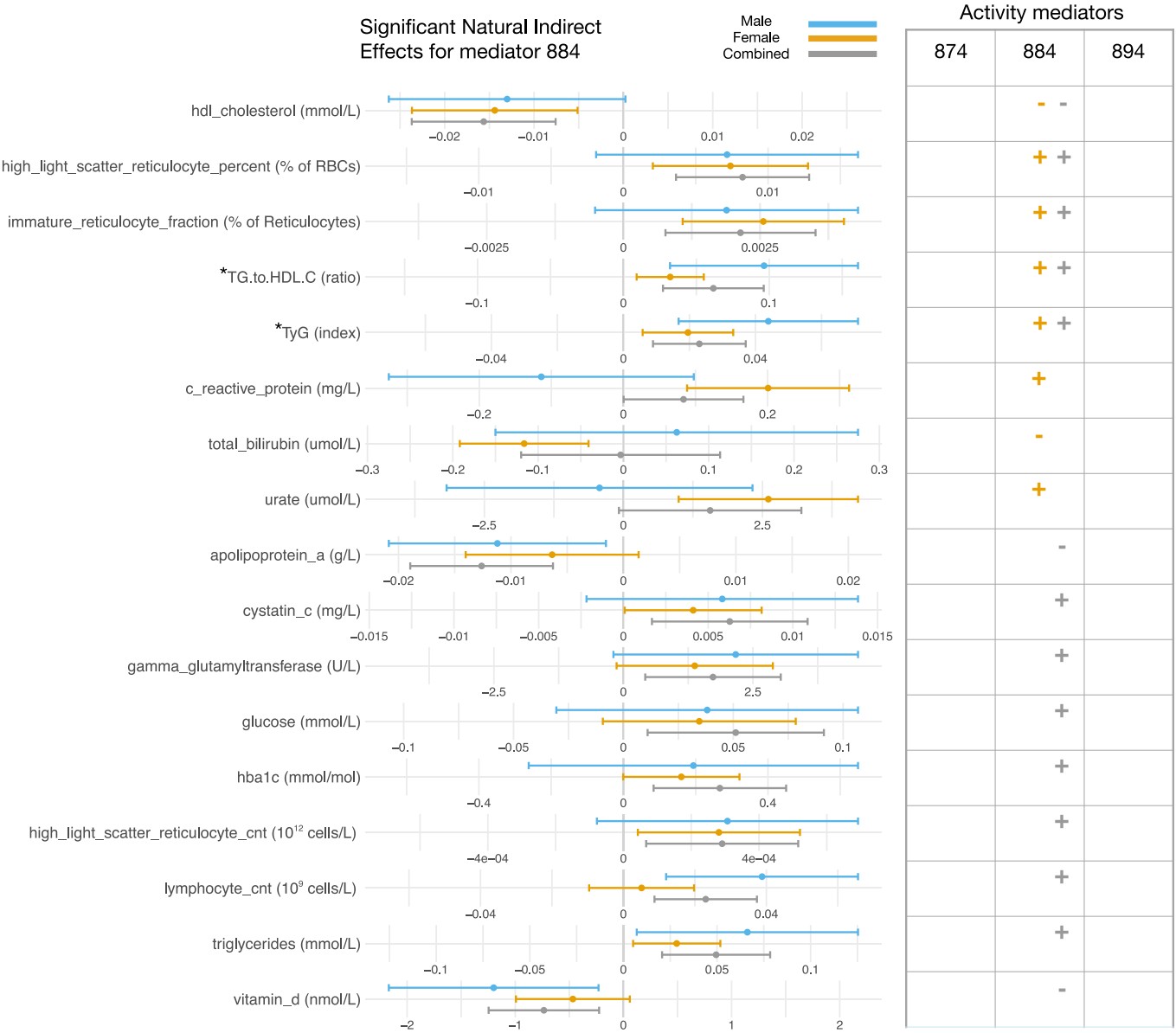

**Figure 3. Associational natural indirect effects (NIE) of ME/CFS on molecular and cellular blood traits.**

The sex-stratified analyses are presented in orange (female) and blue (male). For the combined analysis (grey), sex is additionally taken as a confounder. All traits that are significant for UKB mediator 884 are shown. This is the mediator with the largest number of significant indirect effects. UKB mediator 874 "Duration of walks" has a single significant NIE (mean_corpuscular_haemoglobin} for females) (FDR <0.05), whereas UKB mediator 894 "Duration of moderate activity"' has no significant NIEs (FDR <0.05). Effect sizes are plotted for UKB mediator 884 "Number of days/week of moderate physical activity", for significant estimates (FDR <0.05). Error bars indicate 95% confidence intervals and the central point represents the population average estimate. Note that the scale and unit of measurement for each trait (x axis) are different. Significant results (FDR <0.05) for mediator 884 are indicated by "+" for positive effects and "−" for negative effects. Where there is no symbol, the effect was not significant. Blood trait names are as they appear in the UKB showcase, aside from TyG and TG-to-HDL-C ratio (indicated with *), which are composite measures of other blood traits. Full results can be found in Datasets EV2 and EV3. Sample sizes of each analysis are in EV4.

significant in the PEM analysis, only 9 were significant in the non-PEM analysis. Five blood traits not previously significant (i.e., not shown in Fig. 2A) were significant only in the PEM analysis, and not in the non-PEM analysis: albumin, direct bilirubin, eosinophil percentage, haemoglobin concentration and reticulocyte count. Notably, two biomarkers of chronic inflammation—cystatin C and C-reactive protein—are significant only in the PEM analysis, and not in the non-PEM analysis. We conclude that UKB individuals with PEM have stronger ME/CFS biomarker differences than those without PEM. The full results of the analyses on the combined, male, and female cohorts can be found in Dataset EV11.

## Replication on the All of Us cohort

To seek replication of the UKB results, we analysed data from the All of Us (AoU) Research Program which aims to collect

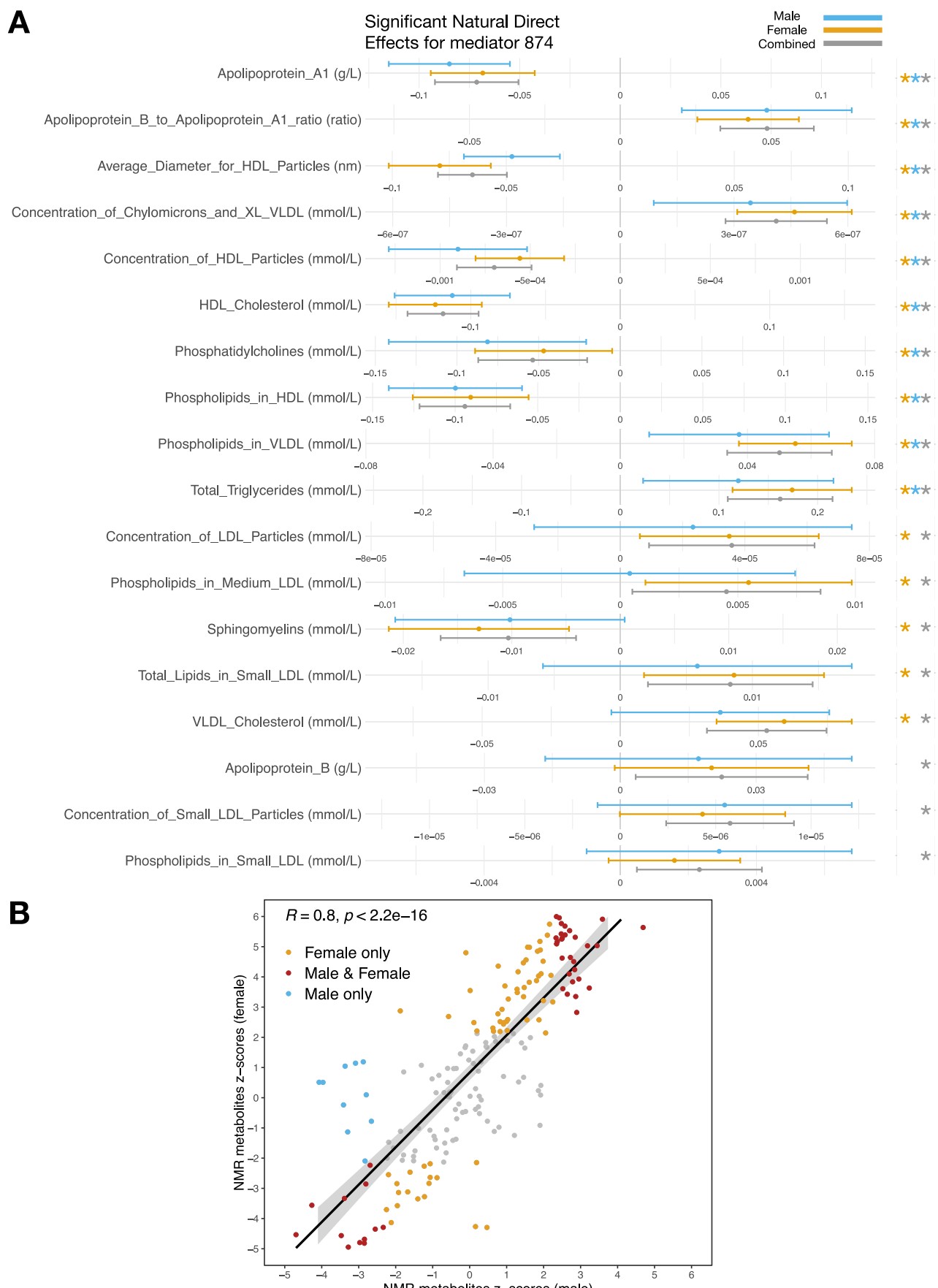

**Figure 4. Associational natural direct effects (NDE) of ME/CFS on NMR metabolites.**

(A) The sex-stratified analyses are presented in orange (female) and blue (male). For the combined analysis (grey), sex is additionally taken as a confounder. Eighteen of 184 traits are shown; results for all traits are provided in Dataset EV5. Effect sizes are plotted for mediator 874 "Duration of walks" for significant estimates (FDR <0.05). Error bars indicate 95% confidence intervals and the central point represents the population average estimate. Note that the scale and unit of measurement (x axis) are different for each metabolite. Asterisks (right) indicate effects that are significant (FDR <0.05). Where there is no asterisk, the effect was not significant. There were no discordant results across the three analyses. All NMR metabolite names are as they appear in the UKB showcase. (B) NMR NDE values are strongly concordant between the two sexes. Shown are per-metabolite z-scores for males (x axis) and females (y axis). Z-scores are the NDE divided by its estimation error. The Pearson correlation is 0.8 and significant ($P = 4.0 \times 10^{-44}$). Red dots indicate metabolites that are significant in both males and females (FDR <0.05). Yellow and blue dots represent metabolites that are significant in females only and males only, respectively (FDR <0.05). Grey dots are significant in neither. Full results can be found in Dataset EV5. Sample sizes of each analysis are in Dataset EV4.

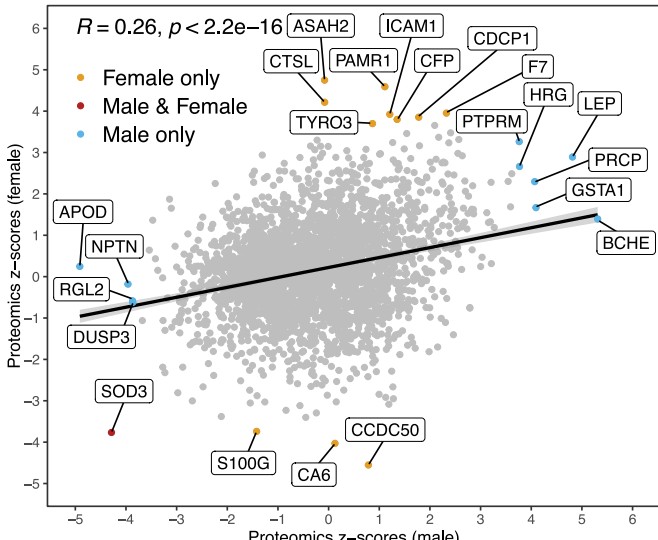

**Figure 5. Protein NDE z-scores, males (x axis) and females (y axis).**

Z-scores are the NDE divided by its estimation error. The Pearson correlation is 0.26 and significant ($P = 5.1 \times 10^{-45}$). The red dot represents the single protein (SOD3) that is significant in both males and females (FDR <0.05). Yellow and blue dots indicate proteins that are significant in females only and in males only, respectively (FDR <0.05). Grey dots show proteins that are significant in neither (i.e., FDR ≥0.05). Full results can be found in Dataset EV6. Sample sizes of each analysis are in Dataset EV4.

health-related data from at least one million individuals across the United States (Kozlowski et al, 2024). Using the AoU Controlled Tier v8 database, we defined 903 ME/CFS cases and 75,943 controls, and tested 12 blood traits (glucose, triglyceride, CRP, AST, ALT, ALP, HDL-C, GGT, HbA1c, leucocyte count, neutrophil count, urea) and 2 composite blood traits (TyG and TG-to-HDL-C ratio) using a 20-fold cross-validation (CV) SL ("Methods"), with stratified CV following SL best practices (Phillips et al, 2023). Of the 14 blood traits tested, 9 were significant in the AoU cohort (FDR <0.05; Dataset EV12; Fig. 8B) with the same direction of effect seen in the UKB.

# Discussion

Our results reveal 511 blood-based biomarkers whose levels differ significantly between people with ME/CFS and those without ME/CFS (Fig. EV2A). Our approach decomposed the total effect of

ME/CFS on blood traits into two components: (1) the indirect effect of ME/CFS on these traits via activity, and (2) the direct effect through all other paths, not mediated via activity. We do not claim causality for our estimates, because the assumptions of no unmeasured confounding may be violated. Nevertheless, any "causal gap", the difference between our estimates and any underlying causal estimand, cannot be due to age and sex, as we account for these factors. Our findings constitute differences in population estimates of blood biomarkers between case and control populations and do not provide individual-level predictions of caseness based on biomarker values. However, our results can be used for variable selection in training a prediction model, as long as an independent dataset is used. If the same data is used twice, i.e., both for variable selection and for training a prediction model, the resulting predictions will suffer from selective inference (Taylor and Tibshirani, 2015), with overly optimistic (invalid) prediction scores, and thus will not generalise to new cases.

A recent study also compared metabolic biomarker levels in ME/CFS cases and controls in UKB and found 168 biomarkers significantly associated with ME/CFS (Huang et al, 2024). Our study additionally analysed UKB blood trait and proteomics data, and defined cases differently. Rather than Huang et al's aim to predict individual-level ME/CFS or control status using NMR metabolite measurements and other symptoms, we sought to quantify, as a population average, blood traits, NMR metabolites or proteins that are significantly different between ME/CFS cases versus controls, controlling for age and sex; whether the effect of ME/CFS on these features occurs indirectly via activity levels; and, whether features differ significantly between sexes. Rather than prediction algorithms, we used semi-parametric estimation theory to quantify population averages, closing any causal gap due to age, sex and BMI (Fig. EV5; Dataset EV13).

The large number of discoveries relative to other studies likely reflects our study's substantially higher numbers of cases and controls (Table 1). These large numbers allow many small average effects of ME/CFS status on molecular and cellular traits to be detected. Importantly, and unlike most previous studies, we independently replicated 166 biomarkers in both females and males (TEs; Fig. EV2A). This indicates that our discoveries are both robust and not sex-biased. It thus provides strong evidence for ME/CFS disease pathophysiology being equivalent in both sexes. This is despite sex-bias of ME/CFS with respect to prevalence and onset, comorbidities, symptoms and other features (Bretherick et al, 2023; Faro et al, 2016).

Importantly, these biomarker differences are not explicable by dissimilarities in physical activity: among 3237 NIE estimates we

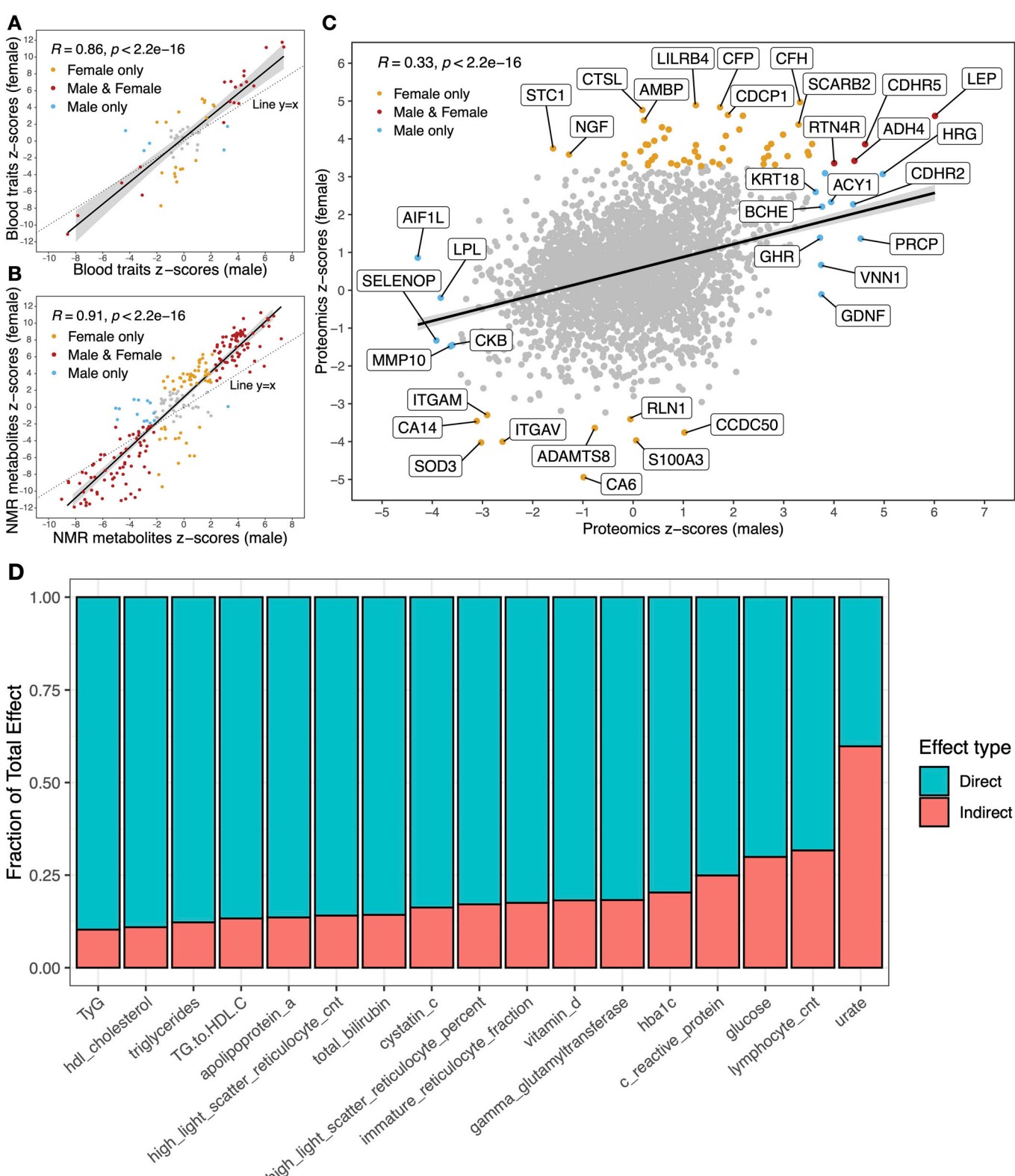

◄ **Figure 6. Concordance of results in male vs female subpopulations in blood traits, NMR metabolites, and proteins, and relative contribution of indirect effects.**

(A) Blood trait total effect z-scores, males (x axis), females (y axis). Z-scores are the TE divided by its estimation error. The Pearson correlation is 0.86 and significant ($P = 4.2 \times 10^{-19}$). The red dots represent 20 blood traits that are significant in both males and females (FDR <0.05). The yellow and blue dots represent blood traits that are significant in females only and males only, respectively (FDR <0.05). The grey dots are significant in neither for FDR <0.05. The $x = y$ line indicates the line of equal z-scores for males and females. In general, in absolute value, the z-scores are higher for females than males. This is to be expected as the sample size is larger for females. (B) Metabolite total effect values are strongly concordant between the two sexes. Shown are per-metabolite z-scores for males (x axis) and females (y axis). Z-scores are the TE divided by its estimation error. The Pearson correlation is 0.91 and significant ($P = 1.5 \times 10^{-97}$). Red dots indicate metabolites that are significant in both males and females (FDR <0.05). Yellow and blue dots represent metabolites that are significant in females only and males only, respectively (FDR < 0.05). Grey dots are significant in neither. (C) Proteins' total effect z-scores, males (x axis), females (y axis). Z-scores are the TE divided by its estimation error. The Pearson correlation is 0.33 and significant ($P = 1.3 \times 10^{-78}$). Red dots represent the proteins (LEP, CDHR5, ADH4, RTN4R) that are significant in both males and females (FDR <0.05). Yellow and blue dots represent proteins that are significant in females only and males only, respectively (FDR <0.05). The grey dots are significant in neither for FDR <0.05. (D) Associational Natural Direct Effect (NDE, blue) and Natural Indirect Effect (NIE, red) as a fraction of the total effect for the effect of ME/CFS on molecular and cellular blood traits. The results are presented for male and female combined, for mediator 884 "Number of days/week of moderate physical activity", the only mediator that exhibits indirect effects. Across all 61 blood traits, and the two composite metrics TyG and TG-to-HLD-C ratio, only 1 feature, Urate, has a larger NIE than NDE, for this mediator only. Full results and sample sizes can be found in Datasets EV3–6.

obtained, ME/CFS status was significantly associated with only one trait (Fig. EV2B). Blood traits thus distinguish ME/CFS cases from population controls, but not because of ME/CFS cases' reduced physical activity levels.

What then causes these molecular and cellular changes in blood if not physical activity? Our findings provide strong and replicated evidence for chronic low-level inflammation (elevated CRP and cystatin C levels, and platelet, leucocyte and neutrophil counts), insulin resistance (elevated triglycerides-to-HDL-C ratio, ALT, ALP, GGT and HbA1c) and/or liver disease (elevated ALT, ALP and GGT, and low urea levels) in ME/CFS (Fig. 2A). ME/CFS is thus portrayed by insulin resistance and systemic inflammation, with liver inflammation and dysfunction likely affecting lipid metabolism and the balance between HDL and LDL cholesterol. To our knowledge, the overall combination of blood marker changes we observed does not present in any other disease. For example, although primary biliary cholangitis is accompanied by elevated ALP and GGT levels (and post-exertional malaise-like symptoms (Jopson et al, 2016)) it is also marked by high circulating levels of bilirubin rather than the lower levels we observe for ME/CFS (Fig. 2A). Nevertheless, because ME/CFS likely arises from multiple pathomechanisms and we did not further stratify cases, we cannot conclude that our results exclude other diseases from sharing a common aetiology with some ME/CFS cases.

In general, shifts in trait values were modest. Among all 116 significant female- and male-replicated traits (NDEs), 91% had small-to-medium shifts (Cohen's d between 0.2 and 0.5 (Cohen, 2013); Dataset EV14). No trait yielded clear separation in estimated effects between ME/CFS cases and controls, rather trait values overlapped extensively. For example, despite CRP level being significantly elevated in ME/CFS cases (TE analysis: adjusted $P = 2.8 \times 10^{-9}$; both sexes), only 4.8% of female and 2.5% of male ME/CFS cases (versus 2.2% and 1.8% controls, respectively) had CRP levels over 10 mg/L, a moderate elevation that can indicate systemic inflammation in autoimmune disease. Consequently, no single blood trait we analysed will be an effective biomarker for ME.

The major strength of the study is its large and deeply phenotyped cohort who were recruited, and their blood traits measured, using a single protocol. The study also controlled for

potential confounders such as age, sex and physical inactivity. Additional mediators beyond physical activity were not considered as they were not directly relevant to this study's principal hypothesis. The study was limited by the UK Biobank's known healthy volunteer bias (Fry et al, 2017), possibly resulting in few, if any, people with severe ME/CFS symptoms at baseline participating. Participants' diet, medication, smoking, alcohol use and socioeconomic status could be confounders but only if they causally influence ME/CFS status. Future studies could test for the effect of symptom severity on the levels of biomarkers found to be significant in this study. UK Biobank recruited 40–69 year old participants (Fry et al, 2017), an age range when individuals are less likely to have a clinical diagnosis of ME/CFS (Samms and Ponting, 2025a). Our analysis also relies on correct clinical categorisation of ME/CFS disease and participants' recall of it. Our choice of clinically defined UKB cohort is pragmatic given that similarly large cohorts, recruited using international consensus diagnostic criteria (Newton et al, 2010; Nacul et al, 2011; Devasahayam et al, 2012), and with blood trait data, have yet to be created. We note that the list of cellular and molecular measurements in the UK Biobank is not exhaustive. For example, others have investigated potential biomarkers for oxidative stress (Shankar et al, 2024) as well as gut metagenomics, immune-profiling and cytokines (Xiong et al, 2024) which are absent from UKB.

Evidence that there is a large number of replicated and diverse blood biomarkers that differentiate between ME/CFS cases and controls should now dispel any lingering perception that ME/CFS is caused by deconditioning and exercise intolerance (Wessely et al, 1989; Moss-Morris et al, 2013; Sharpe, 1995; White et al, 2011). These findings should also accelerate research into the minimum panel of blood traits required to accurately diagnose ME/CFS in real-world populations. Such a panel would be invaluable for diagnosis, for measuring response to future treatment or drug trials, and potentially for determining the worsening or progression of ME/CFS. Such a panel might also help to determine the distinctions or overlap between ME/CFS and symptomologically similar diseases such as Long Covid and fibromyalgia.

To assist the search for an effective biomarker panel for ME/CFS we provide the full results of this study in Datasets EV2–6.

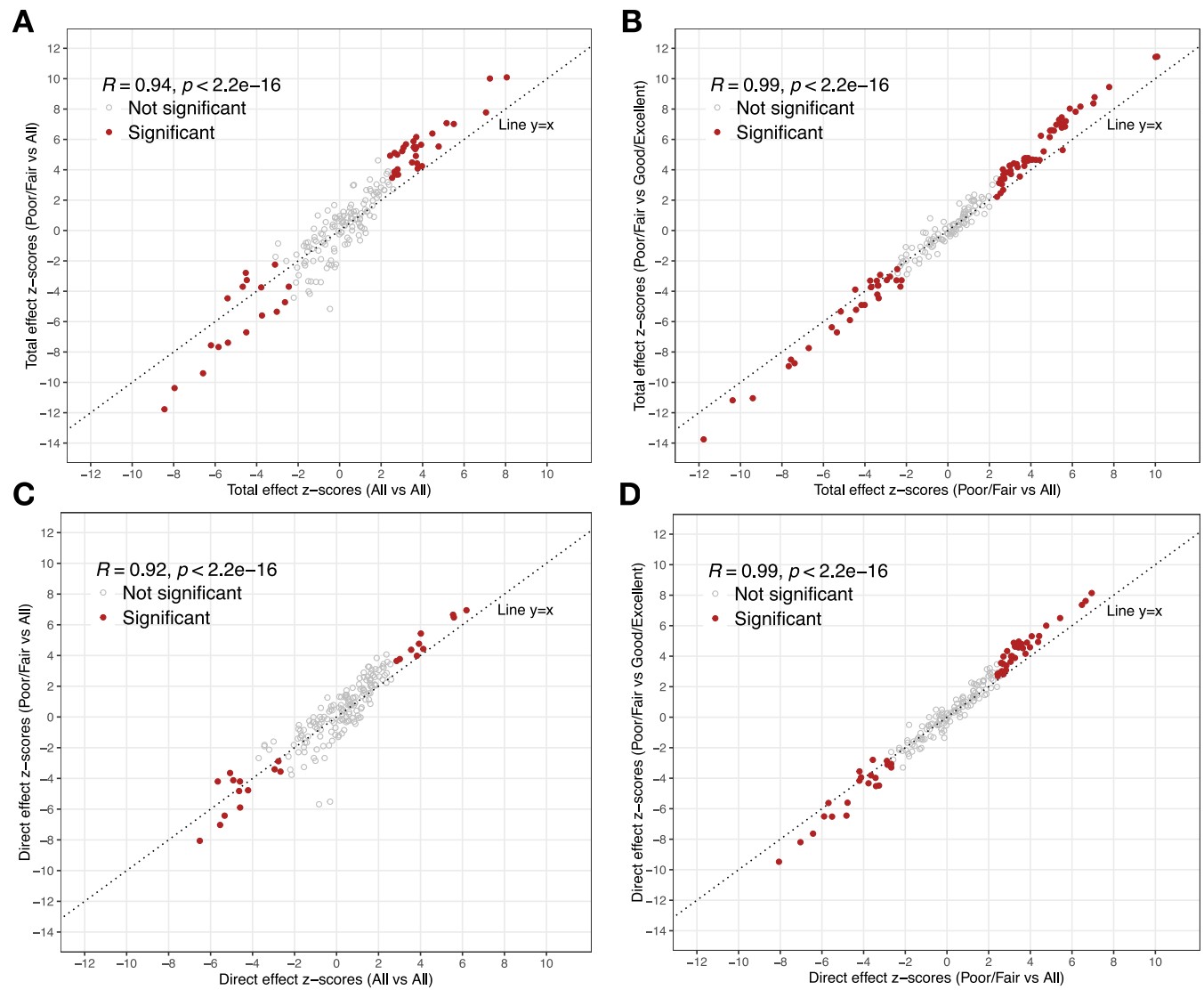

**Figure 7. Total effects and Natural Direct Effects (NDEs) for blood traits become more significant as the stringency of case and control definitions increases.**

(A) Total effect z-scores for 'Poor/Fair' for cases and 'All' (without restricting by health rating (UKB field 2178)) for controls versus z-scores for 'All' for cases and 'All' for controls (without restricting by health rating for cases or controls). The Pearson correlation is 0.94 and significant ($P = 2.7 \times 10^{-89}$). The null hypothesis—that significance does not change for increasing stringency of case or control definition—is represented by the diagonal line. (B) Total effect z-scores for 'Poor/Fair' for cases and 'Good/Excellent' for controls, versus 'Poor/Fair' for cases vs 'All' for controls. The Pearson correlation is 0.99 and significant ($p = 7.5 \times 10^{-165}$). (C, D) As in (A, B) but for NDE. The Pearson correlations are 0.92 and 0.99, respectively, and significant ($P = 6.0 \times 10^{-74}$ and $P = 7.0 \times 10^{-146}$). Full results and sample sizes can be found in Dataset EV10.

# Methods

### Reagents and tools table

| Reagent/resource | Reference or source | Identifier or catalogue number |
|---|---|---|
| **Software** | | |
| *SuperLearner* R package | https://cran.r-project.org/web/packages/SuperLearner/index.html | |
| *sl3* R package | https://github.com/tlverse/sl3 | |

| Reagent/resource | Reference or source | Identifier or catalogue number |
|---|---|---|
| *npcausal* R package | https://github.com/ehkennedy/npcausal | |
| *medoutcon* R package | https://github.com/nhejazi/medoutcon | |
| *haldensify* R package | https://doi.org/10.32614/CRAN.package.haldensify | |
| *speedglm* R package | https://doi.org/10.32614/CRAN.package.speedglm | |

| Reagent/resource | Reference or source | Identifier or catalogue number |
|---|---|---|
| *lightgbm* R package | https://doi.org/10.32614/CRAN.package.lightgbm | |
| *earth* R package | https://doi.org/10.32614/CRAN.package.earth | |
| *ranger* R package | https://doi.org/10.32614/CRAN.package.ranger | |
| *xgboost* R package | https://doi.org/10.32614/CRAN.package.xgboost | |

## UK Biobank ME/CFS data processing

We defined 1455 ME/CFS cases and 131,303 non-ME/CFS control individuals from UKB (Bycroft et al, 2018). For this, our aim was to maximise the study's statistical power by defining cases as those with a ME/CFS clinical diagnosis, and with poor or fair health rating at baseline when blood samples were taken, whilst also seeking to exclude those with other conditions, such as chronic fatigue without ME/CFS. Clinical diagnoses of ME/CFS—particularly more historical diagnoses—will not always have met current clinical guidance (National Institute for Health and Care Excellence (NICE) 2021), including the requirement of PEM and the necessary exclusion of other possible diagnoses. For example, in 2010–2012 studies (Newton et al, 2010; Nacul et al, 2011; Devasahayam et al, 2012) only about half of all participants with a primary diagnosis of ME/CFS in England conformed to the Canadian consensus criteria (Carruthers et al, 2003).

For our study, cases needed to meet each of the four criteria. (i) They self-reported a diagnosis by a doctor of 'Chronic Fatigue Syndrome' (CFS) during a verbal interview at their first visit to a UKB Assessment Centre (UKB field 20002). Note that participants could only self-report CFS, but not ME (or ME/CFS), on that occasion (UK Biobank Verbal Interview stage, version 1.1). (ii) Either they answered "Yes" to the question "Have you ever been told by a doctor that you have myalgic encephalomyelitis/chronic fatigue syndrome?" in the later 'Experience of Pain Questionnaire' (PQ) (2019–2020) (UKB field 120,010), or they did not complete the PQ. This criterion provided supporting evidence for long-term ME/CFS symptoms: there is strong concordance (89%) among those meeting criterion (i) also meeting criterion (ii), despite their responses being approximately 10 years apart. Those providing a discordant response in the two questionnaires were not defined as cases in our study. (iii) In accordance with ME/CFS being debilitating, cases further reported an overall health rating (UKB field 2178) of 'Poor' or 'Fair' at baseline. (iv) They were of known genetic sex.

Population controls did not self-report a CFS diagnosis in any of the 4 visits, answered "No" to the PQ question about a ME/CFS diagnosis, and were not linked to a Primary Care record (CTV3 or Read v2 code, Dataset EV1) of ME/CFS or to the ICD10:G93.3 code ('Postviral fatigue syndrome') in Hospital Inpatient Data. They further reported an overall health rating (UKB field 2178) of 'Good' or 'Excellent' at baseline. UKB participants are older and report healthier lifestyles, higher levels of education and better health relative to the general UK population (Stamatakis et al, 2021; Davis

et al, 2020). UKB assessment at baseline was demanding in time (2–3 h) and energy, including travel to the nearest of 22 centres. These requirements will have diminished the recruitment of people with severe or moderate, or even mild, ME/CFS symptoms. UKB blood samples were acquired and analysed as described previously (Elliott et al, 2008; UK Biobank biochemistry assay quality procedures; UK Biobank companion document for serum biomarker data). On average, the body mass index (BMI; UKB field 21,001) of cases is significantly, but only slightly, higher than the BMI of controls (27.96 ± 4.62 for 386 male cases vs 26.80 ± 3.58 for 55,572 male controls, with $t = 4.92$ and $P = 10^{-5}$, Welch's $t$ test; 27.70 ± 5.83 for 1069 females cases vs 25.82 ± 4.39 for 75,731 female controls with $t = 10.68$ and $P < 10^{-12}$, Welch's $t$ test).

For blood traits, we included two composite markers of insulin resistance: the triglyceride glucose (TyG) index (Che et al, 2023; Si et al, 2021), and TG-to-HDL-C ratio (Cordero and Alegria-Ezquerra, 2009). Note that TyG is normally calculated using fasting levels of triglycerides and plasma glucose (Simental-Mendía et al, 2008), but these are not available from the UK Biobank. The ratio of triglycerides to HDL-cholesterol correlates inversely with the plasma level of small, dense LDL particles.

For NMR metabolomics, we removed individuals whose NMR metabolite measurement has a QC flag indicating irregularities in the measurement, as per UKB category 221.

For each estimator of type TE, NDE and NIE (below), we only considered individuals with the relevant variables measured. Specifically, for TE, we restricted to individuals with measured age, sex and outcome variable. For NDE and NIE, we additionally restricted to individuals with measured mediators of activity. Furthermore, for NDE and NIE, we removed individuals who answered 'do not know' or 'prefer not to answer' to the activity question (UKB datafield 874, 884, or 894).

## Mediation estimators

Causal mediation analysis, concerned with the quantification of the portion of a causal effect of an exposure on an outcome through a particular pathway, has been extensively discussed in the literature (VanderWeele, 2016; Nguyen et al, 2021). The methodologies utilised in this work build upon natural (or pure) mediation estimands (Robins and Greenland, 1992; Pearl, 2001). Strategies for the construction of efficient estimators of non-parametrically defined causal mediation estimands, capable of incorporating machine learning, have been used in a variety of applications. Recent examples include understanding the biological mechanisms by which vaccines causally alter infection risk (Hejazi et al, 2020b; Benkeser et al, 2023; Huang et al, 2023; Hejazi et al, 2023), quantifying the effect of novel pharmacological therapies on substance abuse disorder relapse (Rudolph et al, 2021a; Hejazi et al, 2022c) and the effects of housing vouchers on adolescent development (Rudolph et al, 2021b), and modelling the effects of health disparities on quality of life (Menkir et al, 2024). Here we use state-of-the-art semi-parametric estimation techniques for non-parametric causal mediation analysis (Díaz et al, 2020), implemented in the R package *medoutcon* (Hejazi et al, 2022a, 2022b).

The NDE and NIE are mediational estimands that decompose the average effect (or average treatment effect, ATE) of ME/CFS status on molecular and cellular traits, Eq. (1).

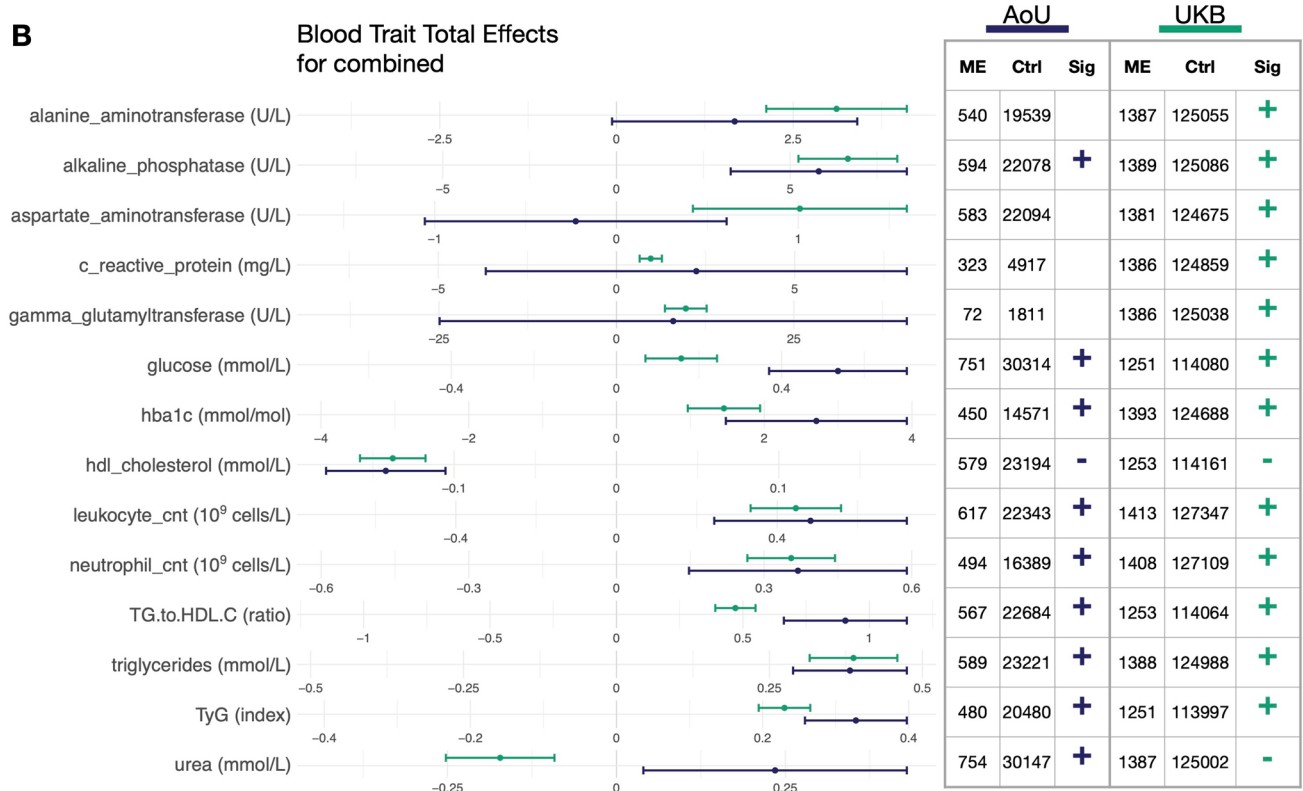

◀

**Figure 8. Comparison of total effects for blood traits among PEM vs non-PEM groups, as well as UKB vs All of Us cohorts.**

(A) Comparison of total effects for blood traits among PEM versus non-PEM ME/CFS samples (both sexes) relative to two independent sets of controls. Identical case and control sample numbers were used in both analyses. Note that there are no significantly opposing results between the two populations. Six blood traits were significant in both analyses (alanine aminotransferase, apolipoprotein a, HDL cholesterol, TG-to-HDL-C ratio, triglycerides and TyG) and are thus replicated. A further 20 traits were significant only in the PEM analysis, and 3 were significant only in the non-PEM analysis. Full results and sample sizes can be found in Dataset EV11. (B) Associational total effects (TE) of ME/CFS on molecular and cellular blood traits in All of Us and UKB, for males and females combined. Age and sex are taken as confounders. Error bars indicate 95% confidence intervals and the central point represents the population average estimate. Note the different scale and unit of measurement used for each trait (x axis). Significant results (FDR <0.05) are indicated by "+" for positive effects and "−" for negative effects. Where there is no symbol shown, the effect was not significant. With the exception of urea, all significant blood traits show concordant directions of effect between AoU and UKB. Full results and sample sizes of each analysis can be found in Dataset EV12.

NDEs involve a comparison of two counterfactual trait outcomes, specifically:

(I)  the level of the trait in a hypothetical scenario where every individual has ME, but rather than allowing ME/CFS to determine the level of activity, we fix their level of activity to the values they would naturally assume if they were not to have ME; and,

(II)  the level of the trait in a hypothetical scenario where every individual is in the control group and their levels of activity are allowed to naturally respond to being in the control group. Comparison of these two trait levels yields a "direct" causal effect that quantifies the effect of ME/CFS on the trait through all paths other than the one mediated by activity.

NIEs involve a comparison of two counterfactual trait outcomes, specifically:

(III)  the level of the trait when every individual has ME/CFS and their levels of activity are allowed to naturally respond to ME; and,

(IV)  the level of the trait in a hypothetical scenario where every individual has ME, but rather than allowing ME/CFS to determine the level of activity, we fix their activity level to the value they would naturally assume if they were not to have ME.

Comparison of these two trait levels yields a causal "indirect" effect that quantifies the impact of ME/CFS on trait through activity (NIE).

Crucially, the counterfactual trait outcomes (I) and (IV) are exactly the same quantity, and this insight gives rise to the "mediation formula" as follows:

$$\underbrace{\mathbb{E}[Y(1) - Y(0)]}_{\text{ATE}} = \mathbb{E}[Y(1, M(1)) - Y(0, M(0))]$$

$$= \underbrace{\underbrace{\mathbb{E}[Y(1, M(0))]}_{\text{I}} - \underbrace{\mathbb{E}[Y(0, M(0))]}_{\text{II}}}_{\text{NDE}}$$

$$+ \underbrace{\underbrace{\mathbb{E}[Y(1, M(1))]}_{\text{III}} - \underbrace{\mathbb{E}[Y(1, M(0))]}_{\text{IV}}}_{\text{NIE}} \qquad (1)$$

where $Y(1)$ and $Y(0)$ are potential outcomes in which an individual does or does not have ME, respectively. Similarly, $Y(1, M(0))$ is the potential outcome of an individual who has ME/CFS and whose mediator takes on the value it would have had if the individual did not have ME/CFS (given in words as (I) and (IV) above). Note also that $Y(1) = Y(1, M(1))$ and $Y(0) = Y(0, M(0))$. The left-hand side of Eq. (1) defines the average treatment effect (ATE) of ME/CFS on

blood trait $Y$, which we refer to as the total effect (TE). The right-hand side of this equation is the sum of the NDE and NIE.

Causal identification is the process of turning a causal quantity we wish to estimate (causal estimand—a functional of unobservable counterfactual data) into a statistical quantity we can estimate from observed data (statistical estimand—a functional of observed data). Causal identification does not require access to any data and is entirely distinct from statistical inference. There are five assumptions required for causal identifiability of Eq. (1):

i.  The Stable Unit Treatment Values Assumption (SUTVA) which includes consistency and no interference between units (Rubin, 1978, 1980);

ii.  exchangeability (unconfoundedness), which is analogous to the randomisation assumption applied to a joint intervention on both the treatment variable (here ME/CFS) and the mediator (here activity);

iii.  treatment positivity, which states that it must be possible to observe any given treatment value (here ME/CFS) across all strata of baseline covariates (age and sex);

iv.  mediator positivity, which states that it must be possible to observe any given mediator value across all strata defined by both treatment (ME/CFS) and baseline covariates (age and sex); and,

v.  Cross-world counterfactual independence $Y(T = t,\ M = m) \perp\!\!\!\perp M(T = t')$ conditional on covariates, which is not empirically verifiable (Robins and Richardson, 2011).

In our case, we do not claim causal identifiability because the assumptions of unconfoundedness (ii) may be violated, as made explicit in Fig. 1A (in red). Nevertheless, we can estimate the NDE and NIE as statistical quantities, knowing that any causal gap will not be due to age or sex, as both of these variables have been taken into account as confounders.

## Super Learner and one-step estimation

We have used semi-parametric efficient estimators to estimate the TE, as well as the mediation effects NDE and NIE (Hejazi et al, 2022b), on multiomic measurements. This estimation procedure consists of an initial Super Learner (SL) (van der Laan et al, 2007) fit to estimate relevant nuisance functions in as flexible a manner as allowed by the available data. This ensures that any model misspecification bias is minimised. We then construct estimates of the NDE and NIE using a one-step bias-correction procedure, which appropriately handles the use of SL for nuisance parameter estimation while also allowing for uncertainty quantification, facilitating the construction of valid Wald-style confidence intervals based on the asymptotic properties of the one-step bias-corrected

estimator (Bickel et al, 1998). The precise specification of these estimators is as follows.

For the total effect, we have used the R package *npcausal* (Kennedy, 2021). This package relies on the *SuperLearner* R package to specify models for fitting nuisance functions. We used:

1. *SL.earth*, an implementation of multivariate adaptive regression splines (Friedman, 1991);
2. *SL.glmnet*, penalised regression with a generalised linear model and hyperparameter $\alpha = 1$, i.e., $L^1$-penalised or Least Absolute Shrinkage and Selection Operator (LASSO) regression, with default tenfold cross-validation;
3. *SL.glm.interaction*, generalised linear model with main terms and two-way interactions;
4. *SL.xgboost*, extreme gradient boosting (XGB) used with default parameters (Chen and Guestrin, 2016).

For the mediation effects NDE and NIE, we used the R package *medoutcon* (Hejazi et al, 2022b). This package instead relies on the *sl3* R package (Coyle et al, 2021), an implementation of the ensemble machine learning algorithm of (van der Laan et al, 2007), to specify models for fitting nuisance functions. We used:

1. *Lrnr_earth*, an implementation of multivariate adaptive regression splines (Friedman, 1991);
2. *Lrnr_glmnet*, penalised linear regression with a generalised linear model and hyperparameters $\alpha = 1$, i.e., $L^1$-penalised or Least Absolute Shrinkage and Selection Operator (LASSO) regression, and default threefold cross-validation;
3. *Lrnr_glm_fast*, a fast implementation of a generalised linear model used with main terms and two-way interactions; and,
4. *Lrnr_lightgbm*, a fast and memory-efficient implementation of extreme gradient boosting (XGB) models from the *lightgbm* R package (Ke et al, 2017), used with default parameters.

The estimation of NDE and NIE relies on the fitting of further nuisance functions for which we have used algorithms, such as the Highly Adaptive Lasso (HAL) (van der Laan, 2017; Hejazi et al, 2020a; Coyle et al, 2023), and parameter specifications recommended by *medoutcon*.

## GO enrichment analysis

We performed Gene Ontology analysis (Ashburner et al, 2000; Aleksander et al, 2023; Sayols, 2023) on the set of significant TE estimates (positive only, negative only, or all combined) obtained from the male, female or combined populations. For the background protein set, we used all 2,923 proteins measured in UKB.

We obtained significant results only for the set of proteins with a significant positive total effect in the female subset of the population at FDR <0.05. The results are presented in Fig. EV3. We used *Rrvgo* (Sayols, 2023) to reduce the redundancy of GO terms.

## Separating those with PEM-like symptoms from those without

Among cases completing the Pain Questionnaire, we identified 297 individuals as having PEM-like symptoms, specifically because they answered (i) 'Yes', (ii) 'Yes', (iii) 'No', and (iv) 'No' to: (i) "Do you have persistent or recurrent tiredness, weariness or fatigue that has lasted at least 6 months?", (ii) "Do you get tired after minimal physical or mental exertion?", (iii) "Does this tiredness, weariness or fatigue go away when you rest?" and (iv) "Is this tiredness, weariness or fatigue happening only because you have been exercising and/or working too much?" These 297 individuals were randomly subsampled to 239 in order to match in number the remaining 239 cases who did not report PEM-like symptoms. With respect to cognitive function, 441 (82%) of our study's participants who completed the Pain Questionnaire reported mild, moderate, or severe cognitive symptoms.

As a further sensitivity analysis of our main total effects analysis of Section "Total effects", we replicated the estimation of the total effect of ME/CFS status on 63 blood biomarkers in the above PEM-like cohort. Specifically, we used as case cohort the 239 individuals with PEM-like symptoms and as control cohort the 239 individuals who did not report PEM-like symptoms but had completed the Pain Questionnaire. For the estimation of the TE, we have used the same specifications and models as described in Section "Super Learner and one-step estimation". In particular, we have used the R package *npcausal* (Kennedy, 2021) and employed semi-parametric efficient one-step estimators to estimate the TE correcting for sex and age.

## All of Us (AoU) cohort

In the AoU Controlled Tier v8 database, there are two groups of relevant standard definitions: (#1) Chronic fatigue syndrome (432738) and (#2) Postviral fatigue syndrome (4202045), each containing four source concepts. In this study we defined AoU ME/CFS cases as participants who have linkage to at least one of the following codes: (#1a) Postviral and related fatigue syndromes (ICD10CM-G93.3), (#1b) Postviral fatigue syndrome (SNOMED-51771007), (#1c) Postviral fatigue syndrome (IC10CM-G93.31), (#1 d) Other post infection and related fatigue syndromes (ICD10CM-G93.39); (#2a) Chronic fatigue syndrome (ICD9CM-780.71), and (#2b) myalgic encephalomyelitis/chronic fatigue syndrome (ICD10CM-G93.32). We did not include (#2c) Chronic fatigue, unspecified (ICD10CM-R53.82) alone, or (#2 d) Chronic fatigue syndrome (SNOMED-52702003) alone, in our inclusion criteria. This is because individuals with codes #2c or #2 d alone are not guaranteed to map to ME/CFS diagnosis codes. In addition, cases needed to have reported a rating of 'Poor' or 'Fair' to the Overall Health Survey question: "In general, would you say your health is?".

Controls were defined as participants who (1) are not linked to #1 or #2 codes, or to 'Post exertional fatigue' (SNOMED-444042007). In addition, controls needed to provide: (i) a rating of 'Excellent', 'Very Good', or 'Good' to the following three questions from the Overall Health Survey: "In general, would you say your health is?", "In general, how would you rate your physical health?" and "In general, how would you rate your mental health, including your mood and ability to think?", (ii) the answer 'Completely' to the question: "To what extent are you able to carry out your everyday activities such as walking, climbing stairs, carrying groceries, or moving a chair?", and (iii) the answer 'None' to the question: "In the past 7 days would you rate your fatigue on average?".

In addition, we restricted case and control individuals to those with recorded sex at birth "Female" or "Male", and a recorded age code. Numbers of cases and controls were 903 and 75,943, respectively.

Unlike UKB, AoU has not consistently collected standardised measurements of activity, and its blood traits have low sample sizes. Consequently, we restricted our replication study to TEs for blood traits, and the combined (male and female) population. We analysed 14 blood traits that were relatively commonly measured in AoU, and that were highly significantly and robustly associated with ME/CFS in the UKB study (Dataset EV7), namely, ALP, ALT, AST, CRP, GGT, glucose, HbA1c, HDL-C, leucocyte count, neutrophil count, triglyceride, and urea, and two composite blood traits, TyG and TG-to-HDL-C ratio. For each of these blood traits, we further winsorised 2.5% extreme values from either end of their combined case and control distribution. When multiple temporal measurements were available per person, we averaged over these measurements to obtain a single value per person. For the two composite quantities, TyG and TG-to-HDL-C, only components measured at the same time were used for their computation, which were then averaged per person across multiple time points to obtain a single value per person. Similar to the UKB study, we controlled for age and sex as confounders and performed an initial SL fit followed by a one-step correction. Given the lower sample sizes in the AoU cohort, a lower statistical power for detecting ME/CFS effects on blood traits was expected. Moreover, estimates were also expected to be more unstable for blood traits with low case numbers due to statistical fluctuations. We thus set up a 20-fold cross-validation (CV) SL, with stratified CV following SL best practices (Phillips et al, 2023).

Full results and sample sizes for each blood trait are provided in Dataset EV12.

### Ethics

UK Biobank has approval from the North West Multi-centre Research Ethics Committee (MREC) as a Research Tissue Bank (RTB) approval (2011, renewed 2016 and 2021). This approval means that researchers do not require separate ethical clearance and can operate under the RTB approval. The basis for consent in UK Biobank rests on participants' explicit and informed consent. UK Biobank uses "legitimate interests" as the primary lawful basis on which to process personal data under the UK GDPR. This study used de-identified data from the All of Us Research Program. The protocol and consent process for the All of Us Research Program were reviewed and approved by the All of Us Institutional Review Board (IRB) under the authority of the National Institutes of Health (NIH). All participants of the All of Us research program provided informed consent for the use of their data in research at the time of enrollment. Additional consent was not required for this secondary analysis. All of Us uses explicit participant informed consent as the lawful basis for the use of All of Us data for biomedical and health research.

## Data availability

This study has not generated data that requires deposition in a public database.

The source data of this paper are collected in the following database record: biostudies:S-SCDT-10_1038-S44321-025-00258-8.

## Peer review information

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

## Acknowledgements

This work was supported by a grant for PhD-level research to GLS from ME Research UK (SCIO charity number SCO36942). This research has been conducted using the UK Biobank Resource under Application Number 76173. This work uses data provided by patients and collected by the NHS as part of their care and support. The authors thank UKB and participants for their many contributions to these projects. Access to the UKB data was funded by the National Institute for Health and Care Research (NIHR) and Medical Research Council (MRC) under grant number MC_PC_20005. The authors gratefully acknowledge All of Us participants for their contributions, without whom this research would not have been possible. We also thank the National Institutes

of Health's All of Us Research Program for making available the participant data examined in this study. AK was supported by a Langmuir Talent Development Fellowship from the Institute of Genetics and Cancer, and a philanthropic donation from Hugh and Josseline Langmuir. SB and AK acknowledge support of the UKRI AI programme, and the Engineering and Physical Sciences Research Council, for CHAI - EPSRC AI hub for Causality in Healthcare AI with real data [grant number EP/Y028856/1]. AMM is supported by the United Kingdom Research and Innovation (grant EP/Y030869/1), UKRI AI Centre for Doctoral Training in Biomedical Innovation at the University of Edinburgh. For the purpose of open access, the author has applied a Creative Commons Attribution (CC BY) license to any author accepted manuscript version arising. SB, AK and CPP are thankful to ME Khamseh for helpful discussions, and to Prof Jo Edwards, Charlie Hillier, Simon McGrath and members of the Science for ME forum for commenting on the initial preprint.

## Author contributions

**Sjoerd Viktor Beentjes**: Conceptualisation; Data curation; Software; Formal analysis; Supervision; Investigation; Visualisation; Methodology; Writing—original draft; Writing—review and editing. **Artur Miralles Méharon**: Data curation; Software; Formal analysis; Investigation. **Julia Kaczmarczyk**: Data curation; Software; Formal analysis; Investigation. **Amanda Cassar**: Data curation; Software; Formal analysis; Investigation. **Gemma Louise Samms**: Data curation; Investigation. **Nima S Hejazi**: Resources; Software; Methodology. **Ava Khamseh**: Conceptualisation; Data curation; Software; Formal analysis; Supervision; Investigation; Methodology; Writing—original draft; Writing—review and editing. **Chris P Ponting**: Conceptualisation; Supervision; Writing—original draft; Writing—review and editing.

Source data underlying figure panels in this paper may have individual authorship assigned. Where available, figure panel/source data authorship is listed in the following database record: biostudies:S-SCDT-10_1038-S44321-025-00258-8.

## Disclosure and competing interests statement

The authors declare no competing interests.

# Expanded View Figures

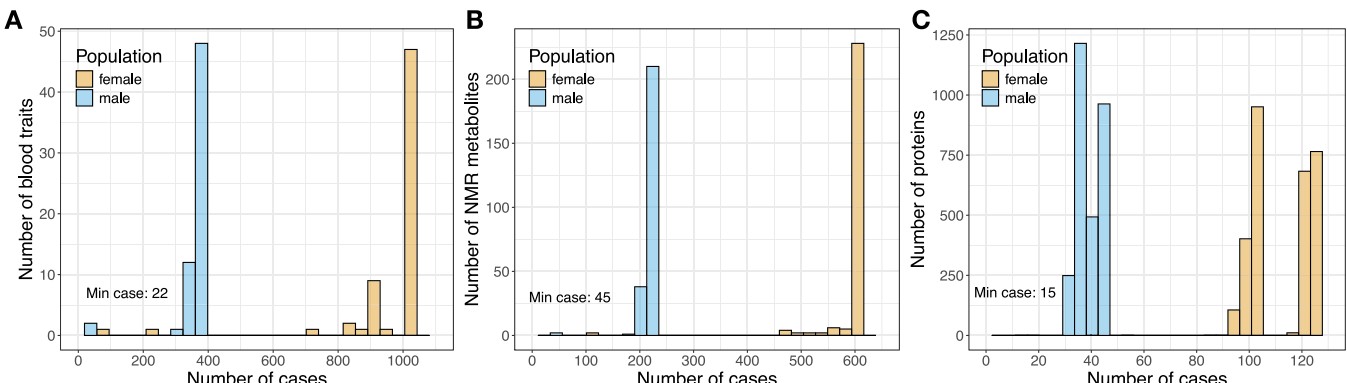

**Figure EV1. ME/CFS sample sizes for males and females, restricting to complete cases (individuals for whom a measurement is available).**

The minimum number of cases is indicated on each plot. (A) Blood traits, (B) NMR metabolites, (C) Proteomics. Neither of the two proteins with case sample size below 30 is significant after FDR correction. Full sample size data is provided as Dataset EV6.

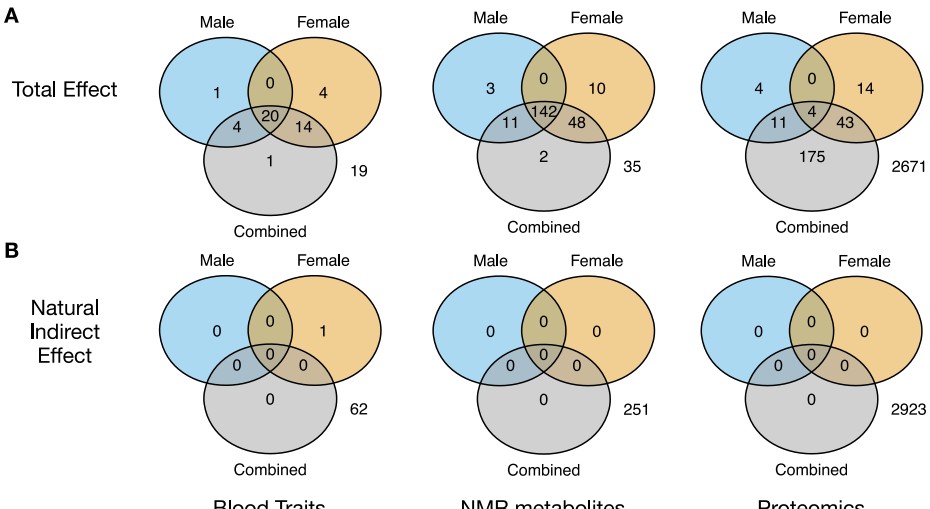

**A** Total Effect

**B** Natural Indirect Effect

Blood Traits · NMR metabolites · Proteomics

**Figure EV2.** Overview of significant associational total effects and natural indirect effects (NIE) in the male, female, and combined populations.

Venn diagrams displaying the number of significant findings in the males, females, combined and their intersection, mediator 874, for (**A**) total effect, and (**B**) NIE.

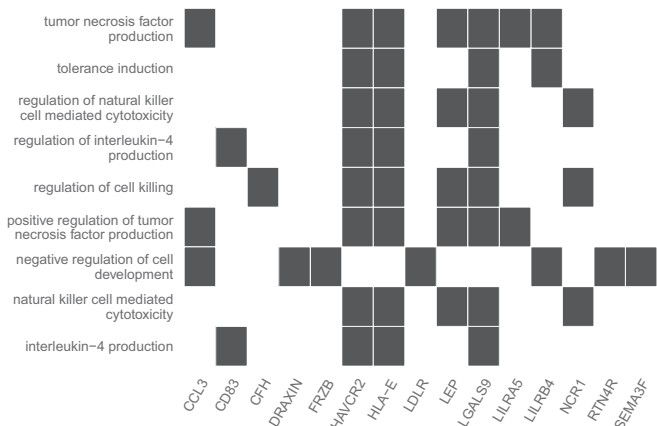

**Figure EV3. GO pathway enrichment (Ashburner et al, 2000) for proteins with a significant positive total effect for ME/CFS vs control, restricted to females only.**

This is the subset with maximal power for GO analysis. All effects are TE, i.e., there are no significant NIE for proteins. We performed a similar pathway GO enrichment analysis for proteins with a significant positive total effect for ME/CFS vs control on the population of males and the combined dataset, as well as all significant negative total effects and all significant total effects on the female, male and combined populations. These resulted in no significant GO term enrichments at FDR <0.05. All measured UKB proteins were used as background for the GO analyses.

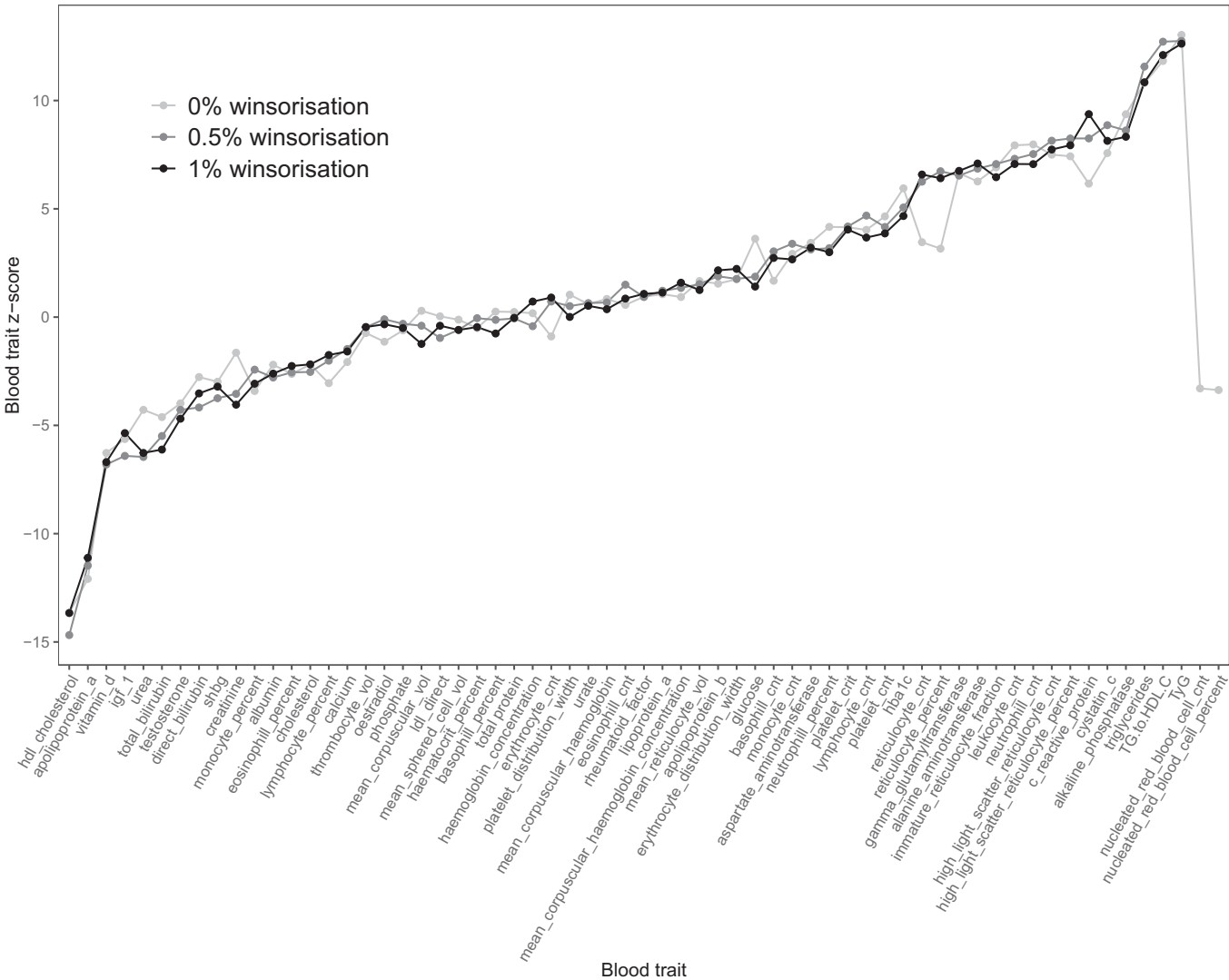

**Figure EV4.  Significant blood traits are robust to winsorisation.**

The points represent total effect z-scores for blood traits in the combined female and male analysis. The three shades of grey represent different degrees of winsorisation of the original data, with cases and controls combined prior to winsorisation. Nucleated red blood cell count and percent are only estimable at 0% winsorisation because for 0.5% winsorisation the number of cases is ≤5. Fib4 and eGFR composite measures were not estimated for 0% winsorisation due to extreme values in control samples (e.g., individuals with platelet counts close to 0). Full results and sample sizes can be found in Dataset EV9.

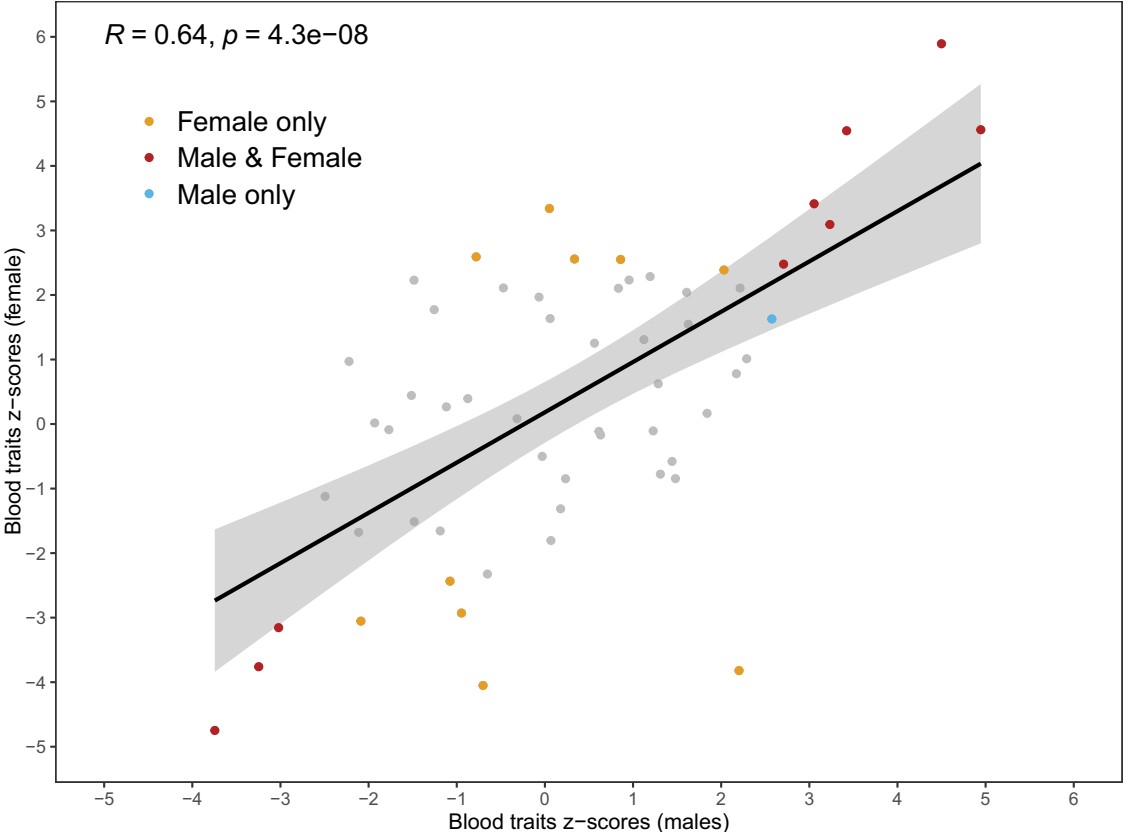

**Figure EV5.  NDE of ME/CFS on blood traits for females and males, for mediator 874, with BMI included as a confounding variable.**

The Pearson correlation is 0.64 and significant ($P = 4.3 \times 10^{-8}$). Data points relating to nucleated red blood cells are not shown due to >90% data missingness. Full results and sample sizes can be found in Dataset EV13.

