## [Peer Review File · EMBO Molecular Medicine]

Replicated blood-based biomarkers for Myalgic Encephalomyelitis not explicable by inactivity

Sjoerd Viktor Beentjes, Artur Miralles Méharon, Julia Kaczmarczyk, Amanda Cassar, Gemma Louise Samms, Nima S. Hejazi, Ava Khamseh, and Chris P. Ponting

Corresponding author(s): Chris Ponting (chris.ponting@ed.ac.uk) , Sjoerd Beentjes (sjoerd.beentjes@ed.ac.uk), Ava Khamseh (ava.khamseh@ed.ac.uk)

Review Timeline:

Submission Date:	17th Oct 24
Editorial Decision:	19th Dec 24
Revision Received:	28th Apr 25
Editorial Decision:	15th May 25
Revision Received:	21st May 25
Accepted:	23rd May 25

Editor: Zeljko Durdevic

Transaction Report:

19th Dec 2024

Dear Prof. Ponting,

Thank you for the submission of your manuscript to EMBO Molecular Medicine, and please accept my apologies for the unusual delay in getting back to you. We have now received feedback from two of the three reviewers who agreed to evaluate your manuscript. As, despite several reminders, the referee #2 will unfortunately not be able to return his/her report in a timely manner, and given that both reviewers provide very similar recommendations, we prefer to make a decision now in order to avoid further delay in the process.

As you will see from their reports pasted below, both referees recognize potential interest of the manuscript but also highlight limitations of the study and raise important concerns that should be addressed in a major revision. During our cross-commenting discussion referee #1 pointed out that there is a very recent publication which also used UK Biobank data to address aspects of ME/CFS, please take this study in consideration when you revise your manuscript. Below I copy the referee #1 additional comment:

"Just after I sent my review of the manuscript, another paper came out, which also used UK Biobank data to address aspects of ME/CFS - PMID: 39592839. As far as I understand, they used some of the same data. However, they asked different questions and used other methods. Therefore, both studies are of high interest, from my point of view. In the other paper, they specifically clarified, near the end of the introduction, that this is "a large heterogeneous ME/CFS cohort". I think a similar clarification would address the problem that a significant proportion may may not meet the consensus diagnostic criteria."

We would welcome the submission of a revised version within three months for further consideration. Please let us know if you require longer to complete the revision.

I look forward to receiving your revised manuscript.

Yours sincerely,

Zeljko Durdevic

We require:

2) Individual production quality figure files as .eps, .tif, .jpg (one file per figure). For guidance, download the 'Figure Guide PDF': (<https://www.embopress.org/page/journal/17574684/authorguide#figureformat>).

3) A .docx formatted letter INCLUDING the reviewers' reports and your detailed point-by-point responses to their comments. As part of the EMBO Press transparent editorial process, the point-by-point response is part of the Review Process File (RPF), which will be published alongside your paper.

4) A complete author checklist, which you can download from our author guidelines (<https://www.embopress.org/page/journal/17574684/authorguide#submissionofrevisions>). Please insert information in the checklist that is also reflected in the manuscript. The completed author checklist will also be part of the RPF.

6) It is mandatory to include a 'Data Availability' section after the Materials and Methods. Before submitting your revision, primary datasets produced in this study need to be deposited in an appropriate public database, and the accession numbers and database listed under 'Data Availability'. Please remember to provide a reviewer password if the datasets are not yet public (see <https://www.embopress.org/page/journal/17574684/authorguide#dataavailability>).

.

- the medical issue you are addressing,

- the results obtained and

- their clinical impact.

12) Author contributions: You will be asked to provide CRediT (Contributor Role Taxonomy) terms in the submission system. These replace a narrative author contribution section in the manuscript.

13) A Conflict of Interest statement should be provided in the main text.

14) Every published paper now includes a 'Synopsis' to further enhance discoverability. Synopses are displayed on the journal webpage and are freely accessible to all readers. They include a short stand first (maximum of 300 characters, including space) as well as 2-5 one-sentences bullet points that summarizes the paper. Please write the bullet points to summarize the key NEW findings. They should be designed to be complementary to the abstract - i.e. not repeat the same text. We encourage inclusion of key acronyms and quantitative information (maximum of 30 words / bullet point). Please use the passive voice. Please attach these in a separate file or send them by email, we will incorporate them accordingly.

15) Include a Reagents and Tools Table as part of the Methods section, which can be downloaded from our author guidelines (<https://www.embopress.org/page/journal/17574684/authorguide#structuredmethods>)

***** Reviewer's comments *****

Referee #1 (Comments on Novelty/Model System for Author):

This study represents an ambitious effort to advance our understanding of ME/CFS using UK Biobank data. The large cohort size significantly strengthens the statistical power of the analyses, which are conducted by a skilled team of bioinformaticians. However, a key limitation is that a substantial portion of the ME/CFS group may not meet consensus diagnostic criteria. This is a challenging issue to address, given the complexity of accurately diagnosing ME/CFS, which warrants rigorous standards for research precision. This concern may be partially mitigated if the authors consider using a broader term, such as "chronic fatigue," for the patient group. I am hopeful that, with careful framing, the study's findings can be responsibly published.

Referee #1 (Remarks for Author):

This is an interesting study aiming to gain new insights about ME/CFS pathology based on UK Biobank (UKB) data. By leveraging advanced statistical methods, the authors quantify two key effects: (1) the direct effect of ME/CFS on molecular and cellular traits, independent of physical activity levels, and (2) the indirect effect mediated by changes in physical activity levels. They analyze three types of blood sample data (blood traits, NMR metabolites, and proteomics) against three separate UKB fields capturing physical activity levels. Notably, they observe that these fields consistently associate with a specific set of biomarkers. While the authors have effectively presented these complex analyses in a relatively well-organized and mostly clear manner, two primary issues remain. First, a significant proportion of the ME/CFS group may not meet consensus diagnostic criteria, which is likely to introduce heterogeneity. Second, additional relevant factors influencing blood biochemistry have not been adequately controlled, which could impact the reliability of the biomarker associations.

Major comments

Comment 1 - the study cohort:

This disease has been referred to by various names over the years, with the most widely recognized terms currently being myalgic encephalomyelitis (ME) and, separately-and less accurately-chronic fatigue syndrome (CFS). Each designation carries a distinct historical background and emphasizes different primary symptoms, which sometimes makes the research literature complicated to read and compare. However, within many healthcare systems, and among healthcare professionals and researchers, the combined term ME/CFS is currently employed for a strictly defined diagnosis.

In the presented study, less specific inclusion criteria were used, based on the data that are available in the UKB:

"Cases self-reported a diagnosis of 'Chronic Fatigue Syndrome' (CFS) in verbal interview at their first visit to a UKB Assessment Centre (UKB field 20002); also, either they answered "Yes" to the question "Have you ever been told by a doctor that you have Myalgic Encephalomyelitis/Chronic Fatigue Syndrome?" in the 'Experience of Pain Questionnaire' (PQ) (2019-2020) (UKB field 120010), or they did not complete the PQ."

It may be problematic to use the "ME/CFS" term on this cohort for several reasons. Hence, there may be a risk of negative impact

on the status of the ME/CFS diagnosis, which probably is not intended, yet a sensitive matter.

Firstly, self-reported diagnosis may capture some genuine cases, but without direct clinical assessment or symptom verification, it risks including individuals who have experienced chronic fatigue but may not meet ME/CFS diagnostic criteria. This can introduce heterogeneity into the ME/CFS cohort.

Next, the definition of ME/CFS cases based on the UKB data falls short of aligning with the stricter consensus diagnostic criteria such as those outlined by IOM (2015), NICE (2021), or CCC (2003), which emphasize specific symptom requirements, such as post-exertional malaise (PEM), unrefreshing sleep, cognitive difficulties, and often orthostatic intolerance. Therefore, the risk is high that the study cohort includes a relatively large proportion of individuals with general chronic fatigue but without the full ME/CFS symptom profile. This is expected to significantly compromise the accuracy of the findings.

If possible, these aspects could be improved for instance by

- including symptom-specific filters: Where possible, ensure that cases have reported PEM, cognitive difficulties, and unrefreshing sleep.

- filter on post-infectious onset

- verifying duration and impact of symptoms: Indicate whether chronicity (symptoms lasting at least six months) and significant functional impairment were considered.

If such evaluation is not possible based on UKB data, it may be challenging to use the "ME/CFS" term. A less specific term, such as "chronic fatigue", would be a better choice.

The performed sensitivity analyses do not deal sufficiently with these aspects.

Comment 2 - Physical activity - exposure or mediator? To address the speculation that avoidance of physical activity lead to ME/CFS - Would it be an idea to test if the outcome becomes different if lack of physical activity is set as the Exposure, and ME/CFS status as the mediator, i.e. switch those to elements in the DAG (Figure 1)?

Comment 3 - BMI? The ME/CFS group has significantly higher BMI than HC (females: 27.70 vs 25.82; males: 27.96 vs 26.80). It is very likely that this has influenced the analysis. For instance, several of the metabolic parameters that were significantly changed in the NDE analysis are known to be influenced both by physical activity and BMI, especially related to blood lipids. Preferably, BMI should be included as a confounder in the analysis. Also, variability in fasting state can be added as a potential influencer contributing to heterogeneity in this context, but this seems difficult to evaluate in the UKB data.

Comment 4 - Biomarkers vs biological phenotypes/changes/mechanisms? The authors use the term "biomarker" on the replicated differentially expressed features between the ME/CFS and HC groups. Although the authors do not claim so, the potential for clinical application is an intrinsic aspect of what makes a biomarker valuable. From my perspective, since the potential value as clinical biomarkers seems limited, it is even more interesting to interpret the findings in terms of biological effects and their possible mechanisms. To this end, especially in this case with an assumed heterogenous patient group, it will be of interest to compare their findings relative to similar studies using strictly defined ME/CFS cohorts. The authors are for instance only scratching the surface when it comes to putting their findings in context of previous reports on metabolic changes, adaptations and phenotypes of ME/CFS, which is a primary topic in the findings. From my perspective, a possible explanation behind the changes in blood lipids is that the underlying pathology causes energy stress and impaired physical performance (e.g. due to hypoperfusion), which next leads to metabolic adaptations.

Minor comments:

- Abbreviations NIE and NDE should be included and explained in Fig 1 and legends

- Fig 3 - show lacking NIE parameter in plot?

Referee #3 (Comments on Novelty/Model System for Author):

While I cannot provide detailed comments on the statistical approach, the aim of this study is highly significant, as no diagnostic tools for ME/CFS are currently available. The challenge lies in the fact that numerous prior studies have explored blood biomarkers, which may dilute the perceived novelty of the findings. Nonetheless, given the large cohort size, this study has the potential to provide a strong foundation for future biomarker research.

Referee #3 (Remarks for Author):

Summary: This study utilized data from the UK Biobank, which includes a large cohort of ME/CFS patients and healthy controls. By integrating components of machine learning, the study aimed to establish a set of blood-based biomarkers with the potential to distinguish ME/CFS patients from healthy controls.

Strength: Large number of ME/CFS patients and healthy controls, with good information on their healthy status and measurables

Limitation: Although the study cohorts are exceptionally large, incorporating data from an additional biobank or independent source would strengthen the findings. However, I acknowledge that this may be challenging due to ethical considerations and time constraints within the study's framework.

I have some comments for the authors to consider:

- 1) While the authors have defined the activities (874, 884, and 894) in their study, do they have additional information on cognitive function or evidence of post-exertional malaise? This information could be valuable, as these are prominent symptoms of ME/CFS.
- 2) While multiple blood traits were associated with chronic inflammation, do the authors have information on cytokine levels?
- 3) I noted that a portion of the proteins identified as significantly different in either individual sex or the entire cohort of ME/CFS patients compared to healthy controls are associated with inflammation. Could the authors comment on the potential links between these proteins and blood traits? Specifically, how might these associations provide insight into the inflammatory processes underlying ME/CFS?
- 4) In addition to the pathways identified by the GO terms, could the authors provide further details on the associated biological processes, cellular components, and molecular functions? Specifically, how do these findings integrate with the identified pathways to enhance our understanding of the proteomic data? If feasible, I would suggest using Ingenuity Pathway Analysis (IPA) for a more comprehensive exploration, though I acknowledge the associated costs might make this analysis challenging to implement.
- 5) Given the inherent risk of overfitting in predictive modelling, particularly with more than 100 variables under consideration, could the authors elaborate on how they address this concern when employing the combination of Super Learner and one-step estimation? Specifically, what strategies or safeguards have been implemented to ensure the generalizability and robustness of the predictive model?
- 6) The authors mentioned that variables not passing QC (e.g., NMR metabolites) were excluded from downstream analysis. I was wondering if:
 - a) There were any missing values for participants across variables such as blood traits, proteomics, or NMR metabolites?
 - b) If missing values were present, how were they handled? For instance, were they imputed, and what approach was used to determine the imputed values?
 - c) Was there a threshold applied for variable inclusion in the analysis, such as excluding variables detected in less than 50% of the combined cohort (healthy and ME/CFS participants)?
- 7) Considering that this study uses clinical data, may I ask whether a human research ethics approval is required for this study?

EMM-2024-20740 Decision

Response Summary:

We thank the reviewers for their thorough and constructive comments. Please find below the summary of additional analyses and changes to the manuscript, followed by a point-by-point response to the reviewers' comments (in blue). Black text below refers to the Editor's/Referees' comments; green text reflects revisions to the current manuscript.

1. Recent study on ME using UK Biobank PMID: 39592839

The Huang et al. study appeared online in November 2024, after we uploaded our manuscript on bioRxiv in August 2024. The reviewer correctly points out that "they used some of the same data. However, they asked different questions and used other methods". We have, of course, now cited this work in the manuscript's Discussion making explicit the differences in questions asked, and the methodologies used:

"A recent study also compared metabolic biomarker levels in ME/CFS cases and controls in UKB and found 168 biomarkers significantly associated with ME/CFS (PMID: 39592839). Our study additionally analysed UKB blood trait and proteomics data, and defined cases differently. Rather than Huang et al.'s aim to predict individual-level ME/CFS or control status using NMR metabolite measurements and other symptoms, we sought to quantify, as a population average, blood traits, NMR metabolites or proteins that are significantly different between ME/CFS cases versus controls, controlling for age and sex; whether the effect of ME/CFS on these features occurs indirectly via activity levels; and, whether features differ significantly between sexes. Rather than prediction algorithms, we used semi-parametric estimation theory to quantify population averages, closing any causal gap due to age, sex and BMI."

As the reviewer requests, we have made further clarifications regarding the heterogeneity of the ME/CFS cohort in UKB (see below).

2. Case definition: This study's ME/CFS cohort required a clinical diagnosis and poor or fair health rating at baseline. Nevertheless, as Referee #1 pointed out, only approximately half of individuals with a ME/CFS clinical diagnosis meet international consensus diagnostic criteria. We now report results from a more restricted ME/CFS cohort, whose definition importantly required postexertional malaise, a key requirement for these consensus criteria. In the revised manuscript's Methods, we now extend our description of case definitions, as well as providing a new section on the definition of cohorts for Fig 9.

3. BMI. We also provide a new analysis (Fig EV5: "NDE of ME/CFS on blood traits for females and males, for mediator 874, with BMI included as a confounding variable") showing a strong

correlation between the original analysis (i.e., BMI not included in the model) and the analysis with BMI included.

4. Reviewer 3 requested that we incorporate “data from an additional biobank or independent source [as this] would strengthen the findings”. To test for replication, we used data from the US-based All of Us (AoU) program, testing 14 traits for 903 ME/CFS cases and 75,943 controls, of which 9 were replicated with the same direction of effect as UKB. It took time to gain access to the AoU platform, install packages, define case/control cohorts and perform the analysis, which explains the time taken for us to respond to reviewers’ comments. These new analyses were mostly performed by Artur Miralles Méharon, who now becomes a new co-author, which has been agreed by all other co-authors.

Additionally, we improved the presentation of Figures 1-8.

Point-by-point response to reviewers’ comments:

Referee #1 (Comments on Novelty/Model System for Author):

This study represents an ambitious effort to advance our understanding of ME/CFS using UK Biobank data. The large cohort size significantly strengthens the statistical power of the analyses, which are conducted by a skilled team of bioinformaticians. However, a key limitation is that a substantial portion of the ME/CFS group may not meet consensus diagnostic criteria. This is a challenging issue to address, given the complexity of accurately diagnosing ME/CFS, which warrants rigorous standards for research precision. This concern may be partially mitigated if the authors consider using a broader term, such as "chronic fatigue," for the patient group. I am hopeful that, with careful framing, the study's findings can be responsibly published.

Referee #1 (Remarks for Author):

This is an interesting study aiming to gain new insights about ME/CFS pathology based on UK Biobank (UKB) data. By leveraging advanced statistical methods, the authors quantify two key effects: (1) the direct effect of ME/CFS on molecular and cellular traits, independent of physical activity levels, and (2) the indirect effect mediated by changes in physical activity levels. They analyze three types of blood sample data (blood traits, NMR metabolites, and proteomics) against three separate UKB fields capturing physical activity levels. Notably, they observe that these fields consistently associate with a specific set of biomarkers. While the authors have effectively presented these complex analyses in a relatively well-organized and mostly clear manner, two primary issues remain. First, a significant proportion of the ME/CFS group may not meet consensus diagnostic criteria, which is likely to introduce heterogeneity. Second,

additional relevant factors influencing blood biochemistry have not been adequately controlled, which could impact the reliability of the biomarker associations.

Major comments

Comment 1 - the study cohort:

This disease has been referred to by various names over the years, with the most widely recognized terms currently being myalgic encephalomyelitis (ME) and, separately-and less accurately-chronic fatigue syndrome (CFS). Each designation carries a distinct historical background and emphasizes different primary symptoms, which sometimes makes the research literature complicated to read and compare. However, within many healthcare systems, and among healthcare professionals and researchers, the combined term ME/CFS is currently employed for a strictly defined diagnosis.

In the presented study, less specific inclusion criteria were used, based on the data that are available in the UKB:

"Cases self-reported a diagnosis of 'Chronic Fatigue Syndrome' (CFS) in verbal interview at their first visit to a UKB Assessment Centre (UKB field 20002); also, either they answered "Yes" to the question "Have you ever been told by a doctor that you have Myalgic Encephalomyelitis/Chronic Fatigue Syndrome?" in the 'Experience of Pain Questionnaire' (PQ) (2019-2020) (UKB field 120010), or they did not complete the PQ."

It may be problematic to use the "ME/CFS" term on this cohort for several reasons. Hence, there may be a risk of negative impact on the status of the ME/CFS diagnosis, which probably is not intended, yet a sensitive matter.

Firstly, self-reported diagnosis may capture some genuine cases, but without direct clinical assessment or symptom verification, it risks including individuals who have experienced chronic fatigue but may not meet ME/CFS diagnostic criteria. This can introduce heterogeneity into the ME/CFS cohort.

Next, the definition of ME/CFS cases based on the UKB data falls short of aligning with the stricter consensus diagnostic criteria such as those outlined by IOM (2015), NICE (2021), or CCC (2003), which emphasize specific symptom requirements, such as post-exertional malaise (PEM), unrefreshing sleep, cognitive difficulties, and often orthostatic intolerance. Therefore, the risk is high that the study cohort includes a relatively large proportion of individuals with general chronic fatigue but without the full ME/CFS symptom profile. This is expected to significantly compromise the accuracy of the findings.

If possible, these aspects could be improved for instance by

- including symptom-specific filters: Where possible, ensure that cases have reported PEM, cognitive difficulties, and unrefreshing sleep.
- filter on post-infectious onset

- verifying duration and impact of symptoms: Indicate whether chronicity (symptoms lasting at least six months) and significant functional impairment were considered.

If such evaluation is not possible based on UKB data, it may be challenging to use the "ME/CFS" term. A less specific term, such as "chronic fatigue", would be a better choice.

The performed sensitivity analyses do not deal sufficiently with these aspects.

Reviewer 1's comment that "self-reported diagnosis may capture some genuine cases, but without direct clinical assessment or symptom verification, it risks including individuals who have experienced chronic fatigue but may not meet ME/CFS diagnostic criteria" is an important point, and we wish to respond in four ways:

(1) First, for clarity, our case definition required individuals to meet ME/CFS diagnostic criteria. This is because they needed to report a clinical diagnosis by a doctor, rather than, for example, a self-diagnosis. This diagnosis would have involved direct clinical assessment and symptom verification and currently requires patients to have experienced 3 months of persistent symptoms. The clinical diagnosis would have been declared at baseline in UKB (the participant selecting that they "have been told by a doctor" that they have other serious illnesses or disabilities, which they then named as Chronic Fatigue Syndrome) or in answer to the Pain Questionnaire (in which they were asked "Have you ever been told by a doctor that you have Myalgic Encephalomyelitis/Chronic Fatigue Syndrome?"). Reviewer 1 refers to "consensus diagnostic criteria", i.e. the Canadian consensus criteria and/or the IOM/NAM criteria. From others' previous research, we expect that approximately half of these cases conform to the Canadian consensus criteria (Newton et al. 2010; Nacul et al., 2011; Devasahayam et al., 2012) at any one time. Note that as ME/CFS is often relapsing-remitting with symptoms arising and varying in severity over time, the same person can meet consensus criteria at one point, but not at another. We understand the need and value of these consensus criteria, yet in our research we do not wish to overlook those diagnosed with ME/CFS who do not meet these criteria at one timepoint. Consequently, we show results relating to the full set of ME/CFS cases. Nevertheless, importantly we also now provide new results (**Fig 9**, see below) relating only to those with an ME/CFS diagnosis who reported, in the PQ, symptoms consistent with post-exertional malaise (PEM) over a 6-month timeframe.

(2) The accuracy of self-reported information has sometimes been questioned. However, in general, self-reported information is known to be reasonably accurate (Smith et al., 2008), especially for certain chronic conditions (Martin et al., 2000). As a chronic condition affecting them daily, people with ME/CFS would be expected to recall their clinical diagnosis accurately.

(3) Notably, rather than over-reporting, self-report health care data is often under-reported by patients (Bhandari & Wagner, 2006). Under-reporting would minimally affect our study's results via mis-classification of controls, in particular because we required control individuals to not be linked to a Primary Care record of ME/CFS or to the ICD10:G93.3 code ('Postviral fatigue syndrome') in Hospital Inpatient Data.

(4) More specifically, a UKB participant's CFS clinical diagnosis self-reported at baseline (when their blood sample was acquired) is supported well by other evidence. Two-thirds (64%) of those who self-reported a CFS diagnosis at baseline have at least one further piece of supporting evidence (Pain Questionnaire ME/CFS response, ICD10:G93.3 code or Primary care record ME/CFS code). This is despite data missingness: e.g., Primary Care records were available for only 45.8% of all UKB participants. When data is available, there is strong concordance: 89% of those self-reporting CFS at baseline also self-report a ME/CFS diagnosis in their response to the Pain Questionnaire. This high level of consistency – in responses approximately 10 years apart – together with a participant's 'Poor' or 'Fair' general health at baseline (as expected for those with an ME/CFS diagnosis), ensured a minimal negative impact on our results from: (i) electronic record coding errors, or (ii) one-off mis-reporting by participants, or (iii) a participant's ME/CFS diagnosis occurring subsequent to collection of their blood sample.

New Post-Exertional Malaise (PEM) Analysis. We now perform a new analysis on those reporting PEM-like symptoms (**Fig 9**). This was made possible because among the ME/CFS cases with available Pain Questionnaire data, 55% reported PEM-like symptoms (see below), answering (i) 'Yes', (ii) 'Yes', (iii) 'No', and (iv) 'No' to: (i) "Do you have persistent or recurrent tiredness, weariness or fatigue that has lasted at least 6 months?", (ii) "Do you get tired after minimal physical or mental exertion?", (iii) "Does this tiredness, weariness or fatigue go away when you rest?" and (iv) "Is this tiredness, weariness or fatigue happening only because you have been exercising and/or working too much?" By separating cases into those reporting or not reporting PEM-like symptoms we find that more biomarkers are significant in the PEM-subset than in the non-PEM subset, and their associations are often stronger (**Fig 9**), despite these two independent analyses being similarly powered.

In the revised submission, we now include these analyses and considerations in the Introduction, Results and Methods.

Introduction: "We analysed molecular and cellular data from up to 1,455 UKB participants, all of whom self-reported a clinical diagnosis of CFS and/or ME/CFS, and from up to 131,303 controls. From other studies, we expect approximately half of these cases to conform to the Canadian consensus criteria (Newton et al., 2010; Nacul et al., 2011; Devasahayam et al., 2012). These criteria require PEM, and indeed, among cases completing the Pain Questionnaire, 55%

report PEM-like symptoms. This allowed our final analysis to compare the effect of ME/CFS status on blood traits for those with and without PEM.”

Results: “No single item of UKB questionnaire or electronic health record data provides incontrovertible evidence of ME/CFS status. Consequently, our choice of case definition was guided by a large fraction (64%) of those self-reporting a CFS diagnosis at baseline having one or more further pieces of evidence (Pain Questionnaire ME/CFS response, or ME/CFS-related codes in General Practice or Hospital Episode Statistics data) in support of their ME/CFS status, despite data missingness (Samms & Ponting, 2025b).”

We have now included a new main **Fig 9** (below) and the following paragraph at the end of the Results:

“Among the ME/CFS cases with available Pain Questionnaire data, 55% reported long-lasting PEM-like symptoms (Methods). This allowed us to compare blood trait results for those with PEM versus those without PEM symptoms. For these two analyses, we used disjoint sets of controls and equal numbers of cases to match statistical power. Results from the two analyses were highly concordant (**Fig. 9**). Nevertheless, whereas 26 biomarkers were significant in the PEM analysis, only 9 were significant in the non-PEM analysis. Five blood traits not previously significant (i.e., not shown in **Fig. 2A**) were significant only in the PEM analysis, and not in the non-PEM analysis: albumin, direct bilirubin, eosinophil percentage, haemoglobin concentration and reticulocyte count. Notably, two biomarkers of chronic inflammation – cystatin C and C-reactive protein – are significant only in the PEM analysis, and not in the non-PEM analysis. We conclude that UKB individuals with PEM have stronger ME/CFS biomarker differences than those without PEM. The full results of the analyses on the combined, male, and female cohorts can be found in Dataset EV11.”

Fig 9: Comparison of total effects for blood traits amongst PEM versus non-PEM ME/CFS samples (both sexes) relative to two independent sets of controls. Identical case and control sample numbers were used in both analyses. Note that there are no significantly opposing results between the two populations. Six blood traits were significant in both analyses (alanine aminotransferase, apolipoprotein a, HDL cholesterol, TG-to-HDL-C ratio, triglycerides and TyG) and are thus replicated. A further 20 traits were significant only in the PEM analysis, and 3 were significant only in the non-PEM analysis.

Methods

4.1. UK Biobank ME/CFS data processing. We defined 1,455 ME/CFS cases and 131,303 non-ME/CFS control individuals from UKB [18]. For this, our aim was to maximise the study's statistical power by defining cases as those with a ME/CFS clinical diagnosis, and with poor or fair health rating at baseline when blood samples were taken, whilst also seeking to exclude those with other conditions, such as chronic fatigue without ME/CFS. Clinical diagnoses of ME/CFS – particularly more historical diagnoses – will not always have met current clinical guidance (NICE Guideline, 2021), including the requirement of PEM and the necessary exclusion of other possible diagnoses. For example, in 2010-12 studies (Newton et al., 2010; Nacul et al., 2011; Devasahayam et al., 2012) only about half of all participants with a primary diagnosis of ME/CFS in England conformed to the Canadian consensus criteria (Carruthers et al., 2003).

For our study, cases needed to meet each of four criteria. (i) They self-reported a diagnosis by a doctor of 'Chronic Fatigue Syndrome' (CFS) during a verbal interview at their first visit to a UKB Assessment Centre (UKB field 20002; Data Ref: UK Biobank, Verbal Interview Stage, 2012). Note that participants could only self-report CFS, but not ME (or ME/CFS), on that occasion (Data Ref: UK Biobank, Verbal Interview Stage, 2012). (ii) Either they answered 'Yes' to the question "Have you ever been told by a doctor that you have Myalgic Encephalomyelitis/Chronic Fatigue Syndrome?" in the later 'Experience of Pain Questionnaire' (PQ) (2019-2020) (UKB field 120010), or they did not complete the PQ. This criterion provided supporting evidence for long-term ME/CFS symptoms: there is strong concordance (89%) among those meeting criterion (i) also meeting criterion (ii), despite their responses being approximately 10 years apart. Those providing a discordant response in the two questionnaires were not defined as cases in our study. (iii) In accordance with ME/CFS being debilitating, cases further reported an overall health rating (UKB field 2178) of 'Poor' or 'Fair' at baseline. (iv) They were of known genetic sex.

The Reviewer says that "It may be problematic to use the "ME/CFS" term on this cohort for several reasons" and suggested that we might use "a broader term, such as "chronic fatigue," for the patient group". For accuracy, we need to use "ME/CFS", rather than "chronic fatigue", because participants self-reported a clinical diagnosis of CFS or ME/CFS, and did not self-report the non-specific term "chronic fatigue" which – if used by us – would misclassify their clinical diagnosis.

The Reviewer says that "Firstly, self-reported diagnosis may capture some genuine cases, but without direct clinical assessment or symptom verification, it risks including individuals who have experienced chronic fatigue but may not meet ME/CFS diagnostic criteria. This can introduce heterogeneity into the ME/CFS cohort." All cases in our study report a clinical diagnosis by a doctor, normally after direct clinical assessment and symptom verification. As we argue above, self-reported information is reasonably accurate (Smith et al., 2008), especially for certain chronic conditions (Martin et al., 2000) such as ME/CFS. Inevitably, our study's definition of UKB cases will have created a heterogeneous cohort. Indeed, any cohort defined using international consensus criteria (e.g., CCC or IOM) would also be heterogeneous due to ME/CFS being multisystemic, often fluctuating or relapsing and remitting, and diagnosed in part by excluding an unlimited set of other diagnoses. Our cohort's definition has benefitted from using multiple and self-consistent data sources: self-report of CFS medical diagnosis at baseline, self-report of ME/CFS medical diagnosis in the Pain Questionnaire, and self-report of General Health at baseline.

It is correct that the ME/CFS case definition we used "falls short of aligning with the stricter consensus diagnostic criteria such as those outlined by IOM (2015), NICE (2021), or CCC (2003)." This was intentional, because no similarly large – and thus similarly powered – ME/CFS cohort with blood trait data yet exists (see our new section on All of Us, below). We now also add a sentence to the Discussion stating this: "Our choice of clinically defined UKB cohort is pragmatic given that similarly large cohorts, recruited using international consensus diagnostic criteria (Carruthers et al., 2003, IOM, 2015, NICE guideline, 2021) and with blood trait data, have yet to be created." We also make this clearer by adding this sentence to the Methods: For example, in

2010-12 studies (Newton et al., 2010; Nacul et al., 2011; Devasahayam et al., 2012) only about half of all participants with a primary diagnosis of ME/CFS in England conformed to the Canadian consensus criteria (Carruthers et al., 2003). Our study design sought to overcome the effects from relatively rare errors in case status using its exceptionally large number of cases. A too narrowly-defined cohort would, we argue, unnecessarily reduce the study's statistical power.

The Reviewer pointed out that the consensus diagnostic criteria require “specific symptom requirements, such as post-exertional malaise (PEM), unrefreshing sleep, cognitive difficulties, and often orthostatic intolerance.” Orthostatic intolerance is an optional symptom for ME/CFS diagnosis, but PEM is required under these criteria. In response to this comment, we performed a new analysis, by defining a subcohort of cases with PEM, and an equal number without PEM-like symptoms. A new Methods section entitled “Separating those with PEM-like symptoms from those without” explains: “Among cases completing the Pain Questionnaire, we identified 297 individuals as having PEM-like symptoms, specifically because they answered (i) ‘Yes’, (ii) ‘Yes’, (iii) ‘No’, and (iv) ‘No’ to: (i) ‘Do you have persistent or recurrent tiredness, weariness or fatigue that has lasted at least 6 months?’, (ii) ‘Do you get tired after minimal physical or mental exertion?’, (iii) ‘Does this tiredness, weariness or fatigue go away when you rest?’ and (iv) ‘Is this tiredness, weariness or fatigue happening only because you have been exercising and/or working too much?’ These 297 individuals were randomly subsampled to 239 in order to match in number the remaining 239 cases who did not report PEM-like symptoms.” Analyses of these subcohorts supported our conclusion that “UKB individuals with PEM have stronger ME/CFS biomarker differences than those without PEM” (Fig 9). We further added the specification and models used in this analysis of total effects for individuals with and without PEM-like symptoms, namely “As a further sensitivity analysis of our main total effects analysis of Section ‘Total Effects’, we replicated the estimation of the total effect of ME/CFS status on 63 blood biomarkers in the above PEM-like cohort. Specifically, we used as case cohort the 239 individuals with PEM-like symptoms and as control cohort the 239 individuals who did not report PEM-like symptoms but had completed the Pain Questionnaire. For the estimation of the TE, we have used the same specifications and models as described in Section ‘Super Learner and one-step estimation’. In particular, we have used the R package *npcausal* and employed semi-parametric efficient one-step estimators to estimate the TE correcting for sex and age.”

The reviewer indicates that “these aspects could be improved for instance by - including symptom-specific filters”, such as on post-infectious onset. Nevertheless, an infectious onset is not required for a ME/CFS diagnosis; approximately one-third of people with ME/CFS do not have an infectious onset. They also requested further information on duration and impact of symptoms. These are now included in the answers to questions quoted in the preceding paragraph and in the revised Methods.

Comment 2 - Physical activity - exposure or mediator? To address the speculation that avoidance of physical activity lead to ME/CFS - Would it be an idea to test if the outcome

becomes different if lack of physical activity is set as the Exposure, and ME/CFS status as the mediator, i.e. switch those to elements in the DAG (Figure 1)?

Thank you for this. It is not possible to address this question without recorded measurements of activity *prior* to ME/CFS status, which are not recorded in UKB. Importantly also, in causal inference the DAG must be fixed *a priori* based on available knowledge about the variables (nodes) and directionalities. The DAG represents substantive knowledge and thus should not be changed between analyses. Also, we note that comparison of estimates from different DAGs based on their significance is causally and statistically invalid (noting the *a priori* null hypothesis).

Comment 3 - BMI? The ME/CFS group has significantly higher BMI than HC (females: 27.70 vs 25.82; males: 27.96 vs 26.80). It is very likely that this has influenced the analysis. For instance, several of the metabolic parameters that were significantly changed in the NDE analysis are known to be influenced both by physical activity and BMI, especially related to blood lipids. Preferably, BMI should be included as a confounder in the analysis. Also, variability in fasting state can be added as a potential influencer contributing to heterogeneity in this context, but this seems difficult to evaluate in the UKB data.

BMI is a risk factor for severe fatigue, but not for ME/CFS (Palacios et al., 2023). BMI can affect activity levels and therefore can be a confounder of the mediator and outcome (blood traits/metabolites/proteins) relationship. For this effect to be taken into account would require pre-ME/CFS measurements of BMI, which are not recorded in the UKB. Nevertheless, as a new sensitivity analysis, we now include the available BMI measured at recruitment and re-ran the blood trait analysis for NDE and NIE. Similar to the result of the original analysis, this yielded no significant NIEs. For NDEs, there were 27 significant blood traits for the combined or male or female populations after controlling for FDR at 5% (**Dataset EV13**). The correlation between the original analysis (i.e., BMI not included in the model) and the analysis with BMI included is presented in the plot below. This plot is now provided in the manuscript as **Fig EV5**.

Figure EV5: **NDE of ME/CFS on blood traits for females and males, for mediator 874, with BMI included as a confounding variable.** Data points relating to nucleated red blood cells are not shown due to >90% data missingness.

Comment 4 - Biomarkers vs biological phenotypes/changes/mechanisms? The authors use the term "biomarker" on the replicated differentially expressed features between the ME/CFS and HC groups. Although the authors do not claim so, the potential for clinical application is an intrinsic aspect of what makes a biomarker valuable. From my perspective, since the potential value as clinical biomarkers seems limited, it is even more interesting to interpret the findings in terms of biological effects and their possible mechanisms. To this end, especially in this case with an assumed heterogenous patient group, it will be of interest to compare their findings relative to similar studies using strictly defined ME/CFS cohorts. The authors are for instance only scratching the surface when it comes to putting their findings in context of previous reports on metabolic changes, adaptations and phenotypes of ME/CFS, which is a primary topic in the findings. From my perspective, a possible explanation behind the changes in blood lipids is that the underlying pathology causes energy stress and impaired physical performance (e.g. due to hypoperfusion), which next leads to metabolic adaptations.

We use the term 'biomarker' to align with the 1998 National Institutes of Health Biomarkers Definitions Working Group: "a characteristic that is objectively measured and evaluated as an indicator of normal biological processes, pathogenic processes, or pharmacologic responses to a therapeutic intervention" (Biomarkers Definitions Working Group, 2001). Our new **Fig 9**

shows results for the more strictly-defined ME/CFS cohorts, i.e. for the PEM ME/CFS comparison.

With respect to metabolic changes, adaptations and phenotypes it would be overly speculative, in our view, to interpret the findings in terms of biological effects and their possible pathomechanisms. This is because it is unknown whether a biomarker level reflects disease cause or downstream consequence. Rather, we highlighted when biomarker results were consistent with previous findings. We do now observe, however, from our new analysis that “two biomarkers of chronic inflammation – cystatin C and C-reactive protein – are significant only in the PEM analysis, and not in the non-PEM analysis” (Results).

Minor comments:

- Abbreviations NIE and NDE should be included and explained in Fig 1 and legends

Thank you, these have now been added.

- Fig 3 - show lacking NIE parameter in plot?

Only effects that remain significant after FDR control (for mediator 884) are plotted. However, we have already provided all estimates (significant or otherwise) in Dataset EV3 (for mediator 874) and Dataset EV2 (for mediators 884 and 894).

Referee #3 (Comments on Novelty/Model System for Author):

While I cannot provide detailed comments on the statistical approach, the aim of this study is highly significant, as no diagnostic tools for ME/CFS are currently available. The challenge lies in the fact that numerous prior studies have explored blood biomarkers, which may dilute the perceived novelty of the findings. Nonetheless, given the large cohort size, this study has the potential to provide a strong foundation for future biomarker research.

Referee #3 (Remarks for Author):

Summary: This study utilized data from the UK Biobank, which includes a large cohort of ME/CFS patients and healthy controls. By integrating components of machine learning, the study aimed to establish a set of blood-based biomarkers with the potential to distinguish ME/CFS patients from healthy controls.

Strength: Large number of ME/CFS patients and healthy controls, with good information on their healthy status and measurables

Limitation: Although the study cohorts are exceptionally large, incorporating data from an additional biobank or independent source would strengthen the findings. However, I acknowledge that this may be challenging due to ethical considerations and time constraints within the study's framework.

In a new Results section “Replication on the All of Us cohort” we now discuss our replication study using this US-based cohort: “Using the AoU Controlled Tier v8 database, we defined 903 ME/CFS cases and 75,943 controls, and tested 12 blood traits (glucose, triglyceride, CRP, AST, ALT, ALP, HDL-C, GGT, HbA1c, leukocyte count, neutrophil count, urea) and 2 composite blood traits (TyG and TG-to-HDL-C ratio) using a 20-fold cross validation (CV) SL (Methods), with stratified CV following SL best practices (Phillips et al, 2023). Of the 14 blood traits tested, 9 were significant in the AoU cohort (FDR <0.05; Dataset EV12; Fig 10) with the same direction of effect seen in the UKB.” The new Fig 10 is provided below for convenience. The corresponding methods are presented in a new final Methods section (“All of Us (AoU) cohort”).

Figure 1: Associational total effects (TE) of ME/CFS on molecular and cellular blood traits in All of Us and UKB, for males and females combined. Age and sex are taken as confounders. Error bars indicate 95% confidence intervals. Note the different scale and unit of measurement used for each trait (x-axis). Significant results (FDR <0.05) are indicated by “+” for positive effects and “-” for negative effects. Where there is no symbol shown, the effect was not significant. With the exception of urea, all significant blood traits show concordant directions of effect between AoU and UKB.

I have some comments for the authors to consider:

1) While the authors have defined the activities (874, 884, and 894) in their study, do they have additional information on cognitive function or evidence of post-exertional malaise? This information could be valuable, as these are prominent symptoms of ME/CFS.

Among the complete set of UK Biobank data, only the Pain Questionnaire allows us to identify those with PEM-like symptoms. This questionnaire was completed by 536 (36.8%) of our study's 1,455 participants. Among these 536, 297 (55%) report PEM-like symptoms, and the remaining 239 (45%) do not. This split allowed us to perform the analysis whose results are now shown in **Fig 9**. A new section of the Methods ("Separating those with PEM-like symptoms from those without") describes this approach (please see our response to Referee 1, above).

With respect to cognitive function, 441 (82%) of our study's participants who completed the Pain Questionnaire reported mild, moderate, or severe cognitive symptoms. This is now explicit in the new Methods section: "With respect to cognitive function, 441 (82%) of our study's participants who completed the Pain Questionnaire reported mild, moderate, or severe cognitive symptoms".

2) While multiple blood traits were associated with chronic inflammation, do the authors have information on cytokine levels?

Yes, large numbers of chemokines, interferons, interleukins and other cytokines were studied in the proteomics analysis of our study. Full results were provided as Dataset EV6. In the Main Text we stated that "TNF and IL4 proteins themselves were not significantly altered in abundance."

3) I noted that a portion of the proteins identified as significantly different in either individual sex or the entire cohort of ME/CFS patients compared to healthy controls are associated with inflammation. Could the authors comment on the potential links between these proteins and blood traits? Specifically, how might these associations provide insight into the inflammatory processes underlying ME/CFS?

In general, elevated levels of inflammation markers, such as CRP, are most often caused by infection, or else rheumatologic diseases or neoplasms (Landry et al. 2017). Nevertheless, as in our response to Reviewer 1 (above), we believe that it would be overly speculative to interpret

these findings in terms of biological effects and their possible molecular pathomechanisms. This is because it is unknown whether a biomarker level reflects disease cause or downstream consequence.

4) In addition to the pathways identified by the GO terms, could the authors provide further details on the associated biological processes, cellular components, and molecular functions? Specifically, how do these findings integrate with the identified pathways to enhance our understanding of the proteomic data? If feasible, I would suggest using Ingenuity Pathway Analysis (IPA) for a more comprehensive exploration, though I acknowledge the associated costs might make this analysis challenging to implement.

Many and diverse gene set enrichment analyses could be performed, using different algorithms and data sets. Rather than highlighting any one, or a small subset, unjustifiably we provide the full set of results as Expanded View Datasets, so that any such analysis can be performed by others.

5) Given the inherent risk of overfitting in predictive modelling, particularly with more than 100 variables under consideration, could the authors elaborate on how they address this concern when employing the combination of Super Learner and one-step estimation? Specifically, what strategies or safeguards have been implemented to ensure the generalizability and robustness of the predictive model?

We have not performed any predictive modelling in this study. All results are population estimates for marginal (single) blood traits, controlling for age, sex and mediator variables. The Super Learning procedure can be used for estimation or for prediction, and here it has only been used for estimation; consequently, there is no notion of “held-out” data, as required for prediction studies. Nevertheless, k-fold cross-validation has been performed to select, in a data-driven manner, the best algorithms (in terms of L2 loss) for the initial estimate of outcome as a function of ME/CFS status, mediator, age and sex. The algorithms we used for the SL are described in Methods, including algorithms, such as HAL, with mathematical guarantees to achieve the required convergence rate of $n^{-1/4}$. The one-step bias-correction procedure was then applied to: (i) reduce any remaining model-misspecification bias for the target quantity of interest (TE, NDE or NIE), and (ii) construct the corresponding valid Wald-style 95% confidence intervals. These details are in the Methods.

6) The authors mentioned that variables not passing QC (e.g., NMR metabolites) were excluded from downstream analysis. I was wondering if:

a) There were any missing values for participants across variables such as blood traits,

proteomics, or NMR metabolites?

b) If missing values were present, how were they handled? For instance, were they imputed, and what approach was used to determine the imputed values?

c) Was there a threshold applied for variable inclusion in the analysis, such as excluding variables detected in less than 50% of the combined cohort (healthy and ME/CFS participants)?

There are missing values for individuals for blood traits in UKB, with the greatest number of missingness in blood biochemistries, followed by blood counts, and fewer for NMR metabolites and proteins. Imputation is more relevant for predictive modelling, to avoid loss of samples and/or features. However, as mentioned above, since we are concerned with marginal effect estimation (as opposed to prediction) for each of the blood traits separately, we constrained each (population average) estimation procedure to non-missing values. In the Results, we state: “All estimates restrict to complete cases, removing individuals with missing trait data in that estimate.”

7) Considering that this study uses clinical data, may I ask whether a human research ethics approval is required for this study?

The UK Biobank Research Ethics Committee (REC) approval number is 16/NW/0274. This covers the use of our project, under application number 76173, as stated in the Acknowledgement section. Separate ethics approval is not required by researchers for accessing and analysing UKB data. This information is provided under “Ethical approval”.

15th May 2025

Dear Prof. Ponting,

Thank you for the submission of your revised manuscript to EMBO Molecular Medicine. We have now heard back from the one referee who agreed to evaluate your manuscript. This referee also assessed author responses to concerns raised by referee #3. I am pleased to inform you that we will be able to accept your manuscript pending the following final amendments:

1) Please implement referee's suggestion.

2) Figures: Remove figures from the main manuscript file and move their legends to the end of the file. Please rationalize the number of figures. We note that many figures have only 1-2 panels, so figures with few panels could be merged. Also, please note that a figure should fit one page. Currently, figure 2 is on 2 pages, please correct. Check "Author Guidelines" for more information:

<https://www.embopress.org/page/journal/17574684/authorguide#figureformat>

<https://www.embopress.org/page/journal/17574684/authorguide#expandedview>

3) Tables: Please place Table 1 between the main and the EV figure legends at the end of the manuscript.

4) In the main manuscript file, please do the following:

- Please address all comments suggested by our data editors listed below:

o Figure legends:

1. Please note that the exact p values are not provided in the legends of figures 4B, 5, 6A-C; 8A-D.

2. Please indicate the statistical test used for data analysis in the legends of figures 8A-D; EV5.

3. Please note that information related to n is missing in the legends of figures 2A, 3; 10.

4. Please note that the measure of center for the error bars needs to be defined in the legends of figures 2A, 3, 4A, 10.

- Add up to 5 keywords.

- Add callouts for Fig. 6.

- Remove "data not shown" (p. 16 and 32).

- Remove "Draft Summary".

- In Methods, add statistical paragraph that should reflect all information that you have filled in the Authors Checklist, especially regarding randomization, blinding, replication etc.

- Indicate in legends exact n and exact p values, not a range, along with the statistical test used. To keep the figures "clear"

some authors found providing an Appendix table Sx with all exact p-values preferable. You are welcome to do this if you want to.

- Author contributions: Please remove it from the manuscript and specify author contributions in our submission system. CRediT has replaced the traditional author contributions section because it offers a systematic machine-readable author contributions format that allows for more effective research assessment. You are encouraged to use the free text boxes beneath each contributing author's name to add specific details on the author's contribution. More information is available in our guide to authors:

<https://www.embopress.org/page/journal/17574684/authorguide#authorshipguidelines>

- Move "Ethical approval" and "Consent" to Methods section and place both in the paragraph named "Ethics".

- In data availability please remove the sentence "Researchers can apply for access to UK Biobank or All of Us data via their websites: <https://www.ukbiobank.ac.uk/> and <https://www.researchallofus.org/register/>".

5) Reagent Table: Please remove example table form the file.

6) Datasets: Please remove the file "Captions to EV datasets". The legends of each corresponding dataset in the file is sufficient.

7) Funding: Please make sure that information about all sources of funding are complete in both our submission system and in the manuscript. Currently, P/Y028856/1, Langmuir Talent Development Fellowship and philanthropic donation from Hugh and Josseline Langmuir are missing in our submission system. Please correct.

8) Synopsis

- Synopsis text: Please remove it from the manuscript and upload it as a separate .doc file.

9) As part of the EMBO Publications transparent editorial process initiative (see our Editorial at

<http://embomolmed.embopress.org/content/2/9/329>), EMBO Molecular Medicine will publish online a Review Process File (RPF)

to accompany accepted manuscripts. This file will be published in conjunction with your paper and will include the anonymous referee reports, your point-by-point response and all pertinent correspondence relating to the manuscript. Let us know whether you agree with the publication of the RPF and as here, if you want to remove or not any figures from it prior to publication.

10) Please provide a point-by-point letter INCLUDING my comments as well as the reviewer's reports and your detailed responses (as Word file).

I look forward to reading a new revised version of your manuscript as soon as possible.

Yours sincerely,

Zeljko Durdevic

Zeljko Durdevic
Senior Editor
EMBO Molecular Medicine

*** Instructions to submit your revised manuscript ***

- 1) a .docx formatted version of the manuscript text (including Figure legends and tables)
- 2) Separate figure files*
- 3) supplemental information as Expanded View and/or Appendix. Please carefully check the authors guidelines for formatting Expanded view and Appendix figures and tables at <https://www.embopress.org/page/journal/17574684/authorguide#expandedview>
- 4) a letter INCLUDING the reviewer's reports and your detailed responses to their comments (as Word file).
- 5) The paper explained: EMBO Molecular Medicine articles are accompanied by a summary of the articles to emphasize the major findings in the paper and their medical implications for the non-specialist reader. Please provide a draft summary of your article highlighting
 - the medical issue you are addressing,
 - the results obtained and
 - their clinical impact.This may be edited to ensure that readers understand the significance and context of the research. Please refer to any of our published articles for an example.
- 6) Author contributions: the contribution of every author must be detailed in a separate section.
- 7) EMBO Molecular Medicine now requires a complete author checklist (<https://www.embopress.org/page/journal/17574684/authorguide>) to be submitted with all revised manuscripts. Please use the checklist as guideline for the sort of information we need WITHIN the manuscript. The checklist should only be filled with page numbers where the information can be found. This is particularly important for animal reporting, antibody dilutions (missing) and exact values and n that should be indicated instead of a range.
- 8) Every published paper now includes a 'Synopsis' to further enhance discoverability. Synopses are displayed on the journal webpage and are freely accessible to all readers. They include a short stand first (maximum of 300 characters, including space) as well as 2-5 one sentence bullet points that summarise the paper. Please write the bullet points to summarise the key NEW findings. They should be designed to be complementary to the abstract - i.e. not repeat the same text. We encourage inclusion of key acronyms and quantitative information (maximum of 30 words / bullet point). Please use the passive voice. Please attach these in a separate file or send them by email, we will incorporate them accordingly.

You are also welcome to suggest a striking image or visual abstract to illustrate your article. If you do please provide a jpeg file 550 px-wide x 300-600px high.

9) A Conflict of Interest statement should be provided in the main text

10) Please note that we now mandate that all corresponding authors list an ORCID digital identifier. This takes <90 seconds to complete. We encourage all authors to supply an ORCID identifier, which will be linked to their name for unambiguous name identification.

Currently, our records indicate that the ORCID for your account is 0000-0003-0202-7816.

Link Not Available

11) Include a Reagents and Tools Table as part of the Methods section, which can be downloaded from our author guidelines (<https://www.embopress.org/page/journal/17574684/authorguide#structuredmethods>)

Photos 400-800 DPI

*Additional important information regarding figures and illustrations can be found at

<https://bit.ly/EMBOPressFigurePreparationGuideline>. See also figure legend preparation guidelines:

<https://www.embopress.org/page/journal/17574684/authorguide#figureformat>

***** Reviewer's comments *****

Referee #1 (Comments on Novelty/Model System for Author):

The additional work in the revised version has improved the impact.

Referee #1 (Remarks for Author):

I am very pleased with the authors' responses to my comments. They have done significant additional work to address the points I was critical of, particularly regarding the heterogeneity of the ME/CFS cohort and the inclusion of post-exertional malaise (PEM) in their analyses.

The authors have clarified their case definition and provided new results for a more restricted ME/CFS cohort that includes PEM, which is a key requirement for consensus diagnostic criteria. They have also performed new analyses to show the impact of including BMI as a confounding variable and have incorporated data from the US-based All of Us program to strengthen their findings.

Minor comment - I would like to see a brief mention of the new PEM-related findings in the abstract, as I think these results are important.

Overall, I believe the scientific value and impact of the manuscript have been significantly improved, and I support the publication of the article.

The authors addressed the remaining editorial issues.

23rd May 2025

Dear Prof. Ponting,

We are pleased to inform you that your manuscript is accepted for publication and is now being sent to our publisher to be included in the next available issue of EMBO Molecular Medicine.

Zeljko Durdevic
Senior Editor
EMBO Molecular Medicine
